# Genome-wide interrogation of extracellular vesicle biology using barcoded miRNAs

**Albert Lu[1]\*, Paulina Wawro[1], David W Morgens[2], Fernando Portela[1], Michael C Bassik[2], Suzanne R Pfeffer[1]\***

[1]Department of Biochemistry, Stanford University School of Medicine, Stanford, United States; [2]Department of Genetics, Stanford University School of Medicine, Stanford, United States

**Abstract** Extracellular vesicles mediate transfer of biologically active molecules between neighboring or distant cells, and these vesicles may play important roles in normal physiology and the pathogenesis of multiple disease states including cancer. However, the underlying molecular mechanisms of their biogenesis and release remain unknown. We designed artificially barcoded, exosomal microRNAs (bEXOmiRs) to monitor extracellular vesicle release quantitatively using deep sequencing. We then expressed distinct pairs of CRISPR guide RNAs and bEXOmiRs, enabling identification of genes influencing bEXOmiR secretion from Cas9-edited cells. This approach uncovered genes with unrecognized roles in multivesicular endosome exocytosis, including critical roles for Wnt signaling in extracellular vesicle release regulation. Coupling bEXOmiR reporter analysis with CRISPR-Cas9 screening provides a powerful and unbiased means to study extracellular vesicle biology and for the first time, to associate a nucleic acid tag with individual membrane vesicles.

DOI: https://doi.org/10.7554/eLife.41460.001

\*For correspondence:
alulopez@stanford.edu (AL);
pfeffer@stanford.edu (SRP)

## Introduction

Intercellular communication mediated by extracellular vesicles (EVs) has recently attracted significant attention, and may contribute to diverse physiological processes including immune regulation and cancer metastasis. EVs are comprised of two major classes based upon their mechanism of biogenesis: microvesicles are shed directly from the plasma membrane, while exosomes are formed by intra-lumenal budding during multi-vesicular endosome (MVE) biogenesis (*Colombo et al., 2014*; *van Niel et al., 2018*) (*Figure 1A*). EVs contain a wide range of molecules, including antigens, signaling proteins, lipids, and microRNAs (miRNAs) that appear to induce significant biological changes in recipient cells. EV miRNAs released by both tumor and mesenchymal cells are of particular interest, as they may play important roles in metastasis (*Becker et al., 2016*). EV miRNAs secreted by hypothalamic stem cells may regulate aging in mice (*Zhang et al., 2017*). Moreover, EVs have attracted a great deal of attention due to their potential use in diagnostics and therapeutics (*Becker et al., 2016*). Despite their importance, how miRNAs are specifically packaged and released via the EV pathway is poorly understood. The biological significance of EVs also remains controversial, in part due to a lack of understanding of the molecular mechanisms that regulate their formation and release. Here we describe the development of a novel CRISPR screening platform using artificially barcoded miRNAs (bEXOmiRs) that we applied systematically to identify genes involved in EV biology. We show that EV release is regulated tightly by Glycogen Synthase Kinase three and Wnt signaling, and report new players in MVE exocytosis.

**Figure 1.** Development and assessment of bEXOmiRs as high-throughput reporters of EV release. (**A**) Left, Generic structure of bEXOmiRs before and after processing. bEXOmiRs contain a random 15 nucleotide (nt) (base-paired) sequence followed by a constant exosome-targeting motif (GGAGGAG, 'EXO motif'). Middle, intracellular trafficking of bEXOmiRs before export via EVs (exosomes and/or microvesicles). Once secreted, bEXOmiRs can be detected in purified EV fractions by RT-PCR or next generation sequencing. (**B**) Three example bEXOmiRs (A, B, C; *Figure 1—figure supplement 1A*) were cloned into a mammalian expression vector and transfected into HEK293T cells. bEXOmiR expression in both EV (upper panel) and cellular (bottom panel) fractions was assessed by Stem-Loop RT-PCR. Negative control RT-PCR devoid of RT-primer is shown in lane ∅. Red asterisks indicate non-specific PCR amplification bands. (**C**) HEK293T cells transfected with control or RAB27A siRNA. Fresh media was added 24 hr post-transfection; EVs were then isolated 48 hr later. Semi-quantitative RT-PCR gel images (top) show relative levels of a bEXOmiR reporter in both cellular (Cells) and EV fractions from control and RAB27A siRNA-treated cells from two experiments. Graph shows bEXOmiR level after RAB27A depletion, as quantified by qRT-PCR (n = two independent experiments, EXP#1 and EXP#2). Immunoblots at bottom show depletion of RAB27A; α-Tubulin was a loading control. Marker mobility is shown at left in basepairs (**B** and **C**) top panels) or kilodaltons (**C** bottom panels). (**D**) Reproducibility of 5000 bEXOmiR abundances measured by deep sequencing in isolated EVs from two replicate cultures (REP1 and REP2) of K562 cells previously infected with the 5,000-bEXOmiR test library (see text).

DOI: https://doi.org/10.7554/eLife.41460.002

The following figure supplements are available for figure 1:

**Figure supplement 1.** Characterization of bEXOmiRs in EVs.

DOI: https://doi.org/10.7554/eLife.41460.003

**Figure supplement 2.** Characterization of CD63-positive structures in cells with or without bEXOmiR expression.

DOI: https://doi.org/10.7554/eLife.41460.004

## Results and discussion

### bEXOmiRs as reporters of EV release

To create a high-throughput system that monitors EV release by next generation sequencing in a pooled format, we first designed and engineered artificially barcoded, exosomal microRNAs (bEXO-miRs) that could be expressed, targeted and released from cells in EVs (*Figure 1A*; *Figure 1—figure supplement 1A*). bEXOmiRs were designed based upon miR-30 to include a 15 nucleotide (nt) bar-code within the paired, stem sequences, flanked by EV-targeting (GGAGGAG 'EXO motif') and loop

sequences derived from miR-601 that were reported by Sanchez Madrid and colleagues to enhance miRNA exosomal targeting (*Villarroya-Beltri et al., 2013*) (*Figure 1—figure supplement 1A*). Barcode sequences were designed to retain the base pairing characteristics of endogenous miRNA stem regions. After intracellular processing, mature bEXOmiRs would be expected to contain the 15 nt barcode followed by GGAGGAG (22 nt total).

Upon expression of a few trial lentiviruses encoding a bEXOmiR downstream of a GFP open-reading frame in HEK293T cells, bEXOmiR RNAs were detected in EV fractions by RT-PCR, together with established EV protein markers (*Figure 1B*, *Figure 1—figure supplement 1B–D*). The EV-associated, amplified bEXOmiR sequences were derived from an RNA source since control reactions lacking either the first-strand cDNA synthesis primer or reverse transcriptase failed to yield a product (*Figure 1—figure supplement 1C,D*). bEXOmiRs in isolated EVs were protected significantly from RNase digestion only in the absence of nonionic detergent, and they floated in a sucrose density gradient (*Figure 1—figure supplement 1D and E*), consistent with their being membrane-enclosed. The remaining, RNase-resistant bEXOmiRs are likely to be protected by RNA binding proteins. Moreover, treatment with Bafilomycin A1, which enhances EV release (*Edgar et al., 2016*; *Miao et al., 2015*), greatly increased the abundance of both total EV protein markers and bEXOmiR reporters in isolated EV fractions (*Figure 1—figure supplement 1F,G*). bEXOmiR expression did not change the number or size of CD63-positive structures in K562 cells (*Figure 1—figure supplement 2*). Finally, consistent with previous reports (*Ostrowski et al., 2010*), Rab27A depletion resulted in a ~ 50% reduction of a bEXOmiR reporter in these fractions from HEK293T cells (*Figure 1C*, and see below for K562 cells). Thus, bEXOmiRs can be expressed, processed and exported from cells in EVs using previously characterized pathways.

Next, we designed an initial test library comprised of 5000 different bEXOmiRs. The original bEXOmiR design included miR-30 context sequences flanking the hairpin structure (*Figure 1—figure supplement 1A*). Processing of these sequences in miR-30 relies on a structural bulge created by a single base pair mismatch at the base of the hairpin (*Figure 1—figure supplement 1H*). Although our original design did not include this feature, no significant differences were observed in the EV representation of bEXOmiRs with or without the A-C mismatch; we nevertheless retained this motif to be consistent with previous artificial shRNA designs (*Chang et al., 2013*). Comparison of bEXOmiR abundance in two replicate EV analyses showed excellent concordance ($R^2 = 0.95$; *Figure 1D*); this was also true when comparing two independent or replicate screens as described below (see below Figure 3A, B).

The overall representation of bEXOmiRs in EV fractions suggested that intrinsic differences in barcode sequences to some extent, influenced exosomal targeting, as seen for other miRNAs (*Villarroya-Beltri et al., 2013*; *Shurtleff et al., 2016*; *Treiber et al., 2017*). Some bEXOmiRs were very efficiently detected in EVs while others were not (*Figure 1D*). This does not reflect differences in synthesis, because a comparison of EV and cellular bEXOmiR levels showed very poor correlation with probability of release (see Figure 3 C below). An initial search for sorting motifs within barcode sequences using the MEME algorithm (*Bailey et al., 2015*) was unsuccessful. Despite the limitation of some preferential targeting of individual barcodes, these data demonstrate that bEXOmiRs are suitable reporters for high-throughput monitoring of EV release in a pooled format. In addition, certain RNA sequences dominated the sequencing analyses, compromising sequencing depth. Such bEXOmiR outliers were excluded from subsequent screens, and we also size fractionated RNAs and introduced a hairpin oligonucleotide in our sequencing library preparation to remove a major contaminant that dominated sequencing runs. These measures improved the overall sequencing depth significantly enough to permit us to complete a genome-wide screen using suspension grown cell cultures.

## A CRISPR/Cas9 screen using bEXOmiRs

We next used bEXOmiRs in a pooled, genome-wide CRISPR/Cas9-mediated screening protocol to monitor EV release (*Figure 2A*). Briefly, unique CRISPR single guide (sg)RNAs linked by synthesis with distinct bEXOmiRs were co-expressed; we then determined how knockout of every gene influenced EV release by comparing bEXOmiR barcode abundance in EV fractions isolated from wild type compared with Cas9-edited cells (*Figure 2A*). Our hope was that the use of 10 CRISPR sgRNAs per gene, coupled with 10 different barcodes (one per sgRNA) would overcome any limitation of any individual bEXOmiR barcode EV packaging bias, relative efficiencies of different sgRNAs at

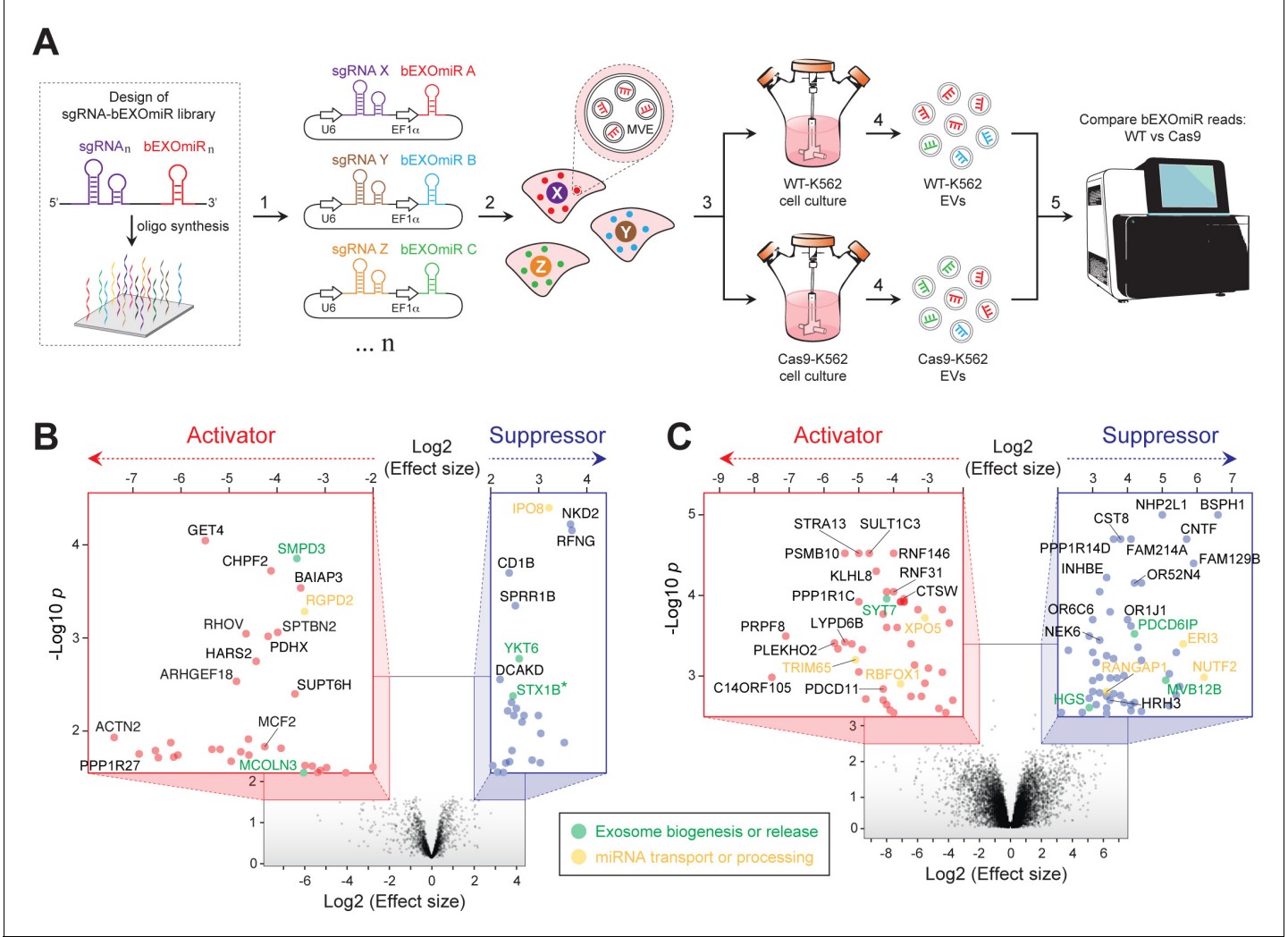

**Figure 2.** Genome-wide CRISPR/Cas9 screen using bEXOmiRs enables systematic interrogation of EV biology. (**A**) Diagram depicting key steps of the screen performed in this study. Oligonucleotides encoding sgRNA-bEXOmiR pairs were designed computationally and then synthesized using solid-phase technology. Next, oligonucleotides were pool-cloned (1) into a Lentivirus vector that drives expression of both sgRNA and bEXOmiR under U6 and EF1α promoters, respectively. WT and Cas9-positive K562 cells were then infected at low MOI (2), such that after infection and selection, each cell expresses only a single sgRNA-bEXOmiR pair (3). This approach enables identification and quantification of both EVs and their respective cell of origin via unique molecular identifiers: barcodes and sgRNAs respectively. sgRNA-bEXOmiR-expressing cells were then grown for 48 hr before collection of culture supernatants from which EVs were purified (4). Finally, barcode abundance was measured, comparing WT and Cas9-expressing cells, by deep sequencing from EV-extracted RNA (*Villarroya-Beltri et al., 2013*). (**B**), Initial pilot screen using a 25,000 bEXOmiR-sgRNA library targeting membrane trafficking, mitochondrial and motility (MMM) genes. Colored insets display zoomed activator (red nodes in red inset) and suppressor (blue nodes in blue inset) hits that passed effect size cutoffs (≤ −2 or ≥+2) and had a -Log p-value>1.5. (**C**), genome-wide (minus MMM sublibrary) screen. Colored insets display zoomed activator and suppressor hits that passed effect size cutoffs (≤ −2 or ≥+2) and had a -Log p-value>2.5. Green labels in (**B**) and (**C**) indicate previously known EV regulators; yellow labels represent genes that regulate miRNA trafficking or processing.

DOI: https://doi.org/10.7554/eLife.41460.005

disrupting gene function, or the real possibility that an sgRNA would decrease a particular bEXO-miR's abundance. In this scenario, gene deletions that decrease extracellular bEXOmiR abundance are considered activators of EV production in WT conditions; in contrast, deletions that increase extracellular bEXOmiR abundance are considered as suppressors of this process.

Two initial, independent pilot screens targeted ~2300 membrane trafficking, motility and mito-chondria-related (MMM) genes, including known EV release-regulators, (eg. SMPD3, RAB27A, MCOLN3, YKT6, etc.) as positive controls and ~200 negative control sgRNAs. As expected, analysis of barcode abundance in EVs isolated from large-scale cultures revealed a gene signature consistent

with previous studies (*Colombo et al., 2014*; *van Niel et al., 2018*; *Miao et al., 2015*; *Ostrowski et al., 2010*; *Kosaka et al., 2010*; *Kosaka et al., 2013*; *Menck et al., 2017*; *Gross et al., 2012*) (*Figure 2B*; *Supplementary file 1*). Among the top hits identified in both initial, independent screens were two known activators: neutral sphingomyelinase (SMPD3), which has been shown to regulate EV-mediated miRNA export (*Kosaka et al., 2010*; *Kosaka et al., 2013*; *Cha et al., 2015*), and mucolipin-3 (MCOLN3), an endolysosomal $Ca^{2+}$ channel important for MVE exocytosis (*Miao et al., 2015*). Additionally, YKT6, a v-SNARE that regulates EV release was identified as a suppressor, as previously reported (*Colombo et al., 2014*; *Gross et al., 2012*). Although endocytosis of EVs by surface proteins might lead to a decrease in EV bEXOmiR abundance, we did not detect a large cluster of endocytosis genes in our screen.

To achieve genome-wide coverage, we performed parallel replicate screens using eight additional sub-libraries (*Figure 2C*, *Supplementary file 1*). Additional known players, including SYT7 and three ESCRT-associated genes (ALIX, HRS and MVB12B) were also confirmed. Interestingly, knockout of any ESCRT-related component consistently increased bEXOmiR abundance in EVs, while knockout of SMPD3 had the opposite phenotype, as previously reported in assays monitoring release of endogenous miRNAs (*Kosaka et al., 2010*). This may reflect distinct EV release pathways. It is also critical to note that knockout of key components specific to one route may artificially increase flux through the other pathway.

Altogether, this analysis (nine duplicated sub-library screens) required >100 liters of cell culture supernatants for EV preparations and associated, large-scale miRNA sequencing runs. *Figure 3A,B* shows a comparison of total bEXOmiR abundance in EV analyses from two replicate screens that similar to our test bEXOmiR (*Figure 1D*), also showed excellent concordance ($R^2 = 0.95$). Shown in *Figure 3C* is a comparison of total bEXOmiR abundance in cells versus EVs. There was essentially no correlation between these pools ($R^2 = 0.13$–$0.17$), consistent with differential sorting of bEXOmiRs into EVs, as observed for endogenous miRNAs. Note that we have not sorted our hits based on EV enrichment in the present analysis. Importantly, comparative analysis of the relative growth of all bEXOmiR expressing cells (determined by genomic sgRNA sequencing) compared with the ability of a given sgRNA to enhance or repress EV levels, revealed no correlation between growth and EV release phenotype (*Figure 3D,E*, *Supplementary file 1*) which would have yielded a diagonal line.

Functional network analysis identified clusters of genes that both positively and negatively regulate specific cellular processes related to EV biogenesis and release, as well as miRNA processing and trafficking (*Figure 4*, *Figure 4—figure supplement 1*). For example, XPO5 and the RAN GTPase activating protein, RANGAP1 gene products regulate miRNA shuttling from the nucleus to the cytoplasm (*Yi et al., 2003*; *Dattilo et al., 2017*), and would be expected to be important for bEXOmiR release. Specific clusters identified include genes needed for regulation of miRNA processing and activity, actin dynamics, Wnt related signaling and trafficking, and ribosome biogenesis (*Figure 4*, *Figure 4—figure supplement 1*). GPCR-related proteins including histamine H3 receptor were also identified (*Figure 2C*, *Figure 4*, *Figure 4—figure supplement 1*). Histamine and neurotransmitter-mediated GPCR signaling have recently been shown to regulate EV release (*Verweij et al., 2018*; *Glebov et al., 2015*). Actin dynamics has also been implicated in the process by which EVs are released from cells (*Li et al., 2012*; *Hoshino et al., 2013*; *Sinha et al., 2016*), and of particular interest in this context was the requirement for ARHGEF18, a guanine nucleotide exchange factor for Rho GTPases (*Zaritsky et al., 2017*). ARHGEF18 regulates actin dynamics in multiple contexts (*Terry et al., 2011*; *Terry et al., 2012*; *Artym et al., 2015*) including invadopodia formation (*Artym et al., 2015*), a process recently proposed to promote EV release (*Hoshino et al., 2013*; *Sinha et al., 2016*).

A number of RNA binding proteins (RBPs) have recently been shown to target miRNAs into EVs, including YBX1, hnRNPA2B1 and SYNCRIP (*Villarroya-Beltri et al., 2013*; *Shurtleff et al., 2016*; *Santangelo et al., 2016*). Although several RNA-binding proteins were uncovered here (i.e. NHP2L1 or LONP1, *Figure 4*, *Figure 4—figure supplement 1*), as well as miRNA-binding proteins XPO5 and PCDC11 (*Treiber et al., 2017*) (*Figure 4*, *Figure 4—figure supplement 1*), we did not find any previously described EV-related RNA binding proteins. This may either be due to their roles in packaging miRNAs that are distinct from the artificial miRNAs to which they were linked in this screen, or to cell type-specific differences (*Villarroya-Beltri et al., 2013*; *Shurtleff et al., 2016*; *Santangelo et al., 2016*).

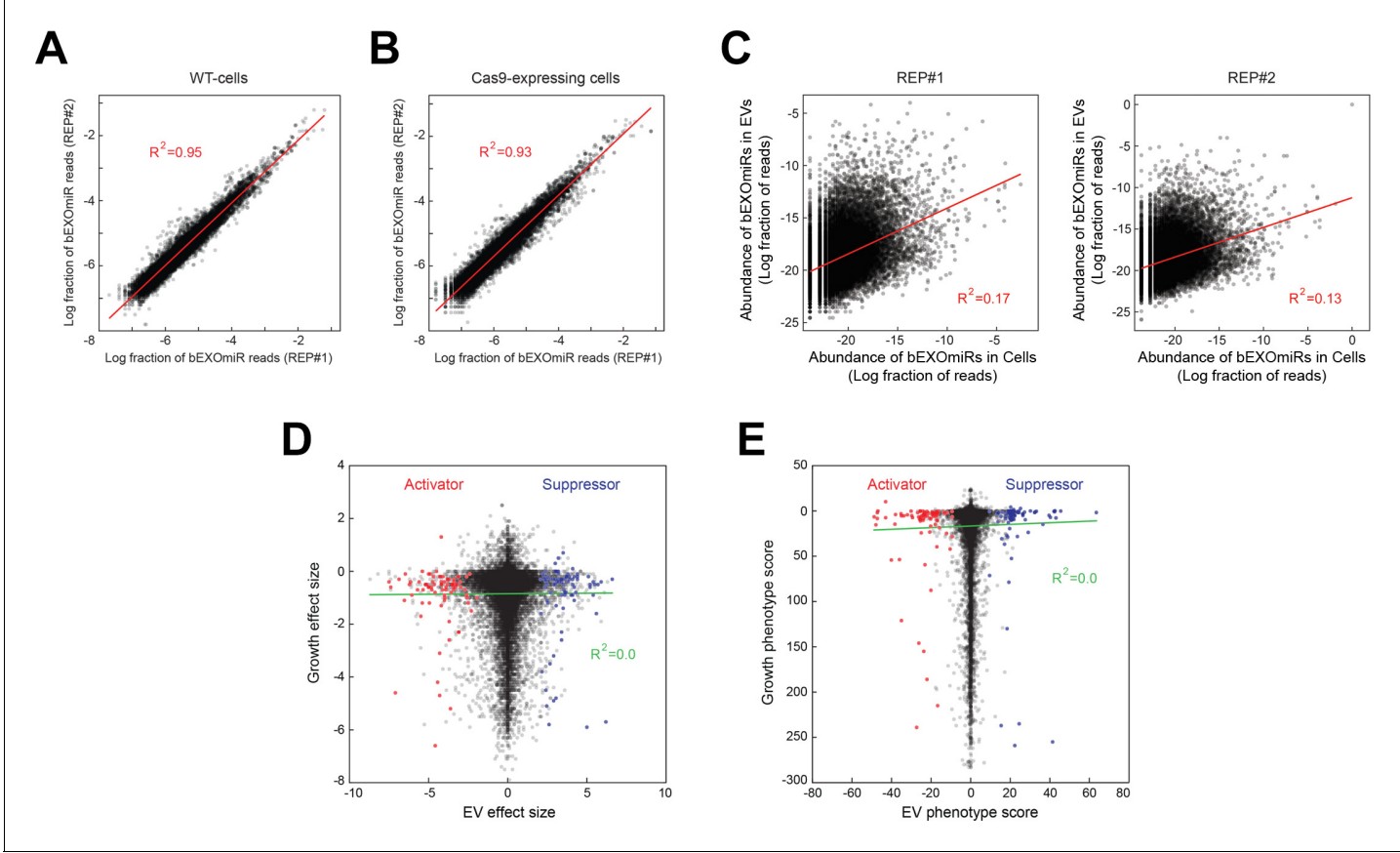

**Figure 3.** Analysis of bEXOmiR EV-targeting. Reproducibility of ~32,000 bEXOmiR abundances measured in EVs isolated from replicated WT- (**A**) or Cas9-positive (**B**) cell cultures (REP#1 and #2). (**C**) Correlation of ~32,000 bEXOmiR abundances in EVs versus intact cells (Cells) from the same WT replicated cultures (left, REP#1; right, REP#2) shown in (**A**). (**D**) Correlation of growth versus EV phenotypes observed in the genome-wide screen. (**E**) Correlation of growth versus EV phenotypic scores calculated using the casTLE algorithm (see Materials and methods) in the genome-wide analysis. Red and blue dots represent identified activator or suppressor hits from the EV generation screen, respectively.

DOI: https://doi.org/10.7554/eLife.41460.006

As mentioned earlier, bEXOmiRs showed significant differences in their efficiency of being detected in EV fractions (*Figure 1D*, *Figure 3A,B*). To explore the impact of such heterogeneity on overall screen outcomes, we tested the ability of different bEXOmiRs to function when paired with the same individual sgRNAs used to target the 100 top hits identified in our initial MMM screen (*Figure 2B*, *Figure 5—figure supplement 2A,B*, *Supplementary file 1*). Although some differences were observed in terms of our ability to detect various players in screens run with different bEXOmiRs/gRNA pairs, we were nevertheless able to re-confirm the findings obtained in the initial screen. Unexpectedly, YKT6, which appeared as a suppressor in the initial screen (*Figure 2B*), was uncovered as a slight activator instead; this may be related to differences in the bEXOmiRs associated with this gene in the two separate analyses. Indeed, YKT6 appears to be required for EV-dependent Wnt release along with STX1A (*Colombo et al., 2014*; *Gross et al., 2012*), a close homologue of STX1B (*Figure 2B*). It is important to note that the phenotype observed relies on the efficiencies of both sgRNAs and their paired, bEXOmiRs: some guides work better than others, and the same applies for bEXOmiRs. This limitation highlights the importance of orthogonal hit-validation experiments. Usually 2–3 guide-bEXOmiR associations supported each hit, and in most cases more than 90% of bEXOmiRs were recovered upon EV fraction sequencing (*Supplementary file 2*).

## Hit validation

The importance of ARHGEF18 in EV biogenesis was confirmed using an orthogonal, Nanostring method to profile the presence of 74 endogenous miRNAs in EVs produced by polyclonal human

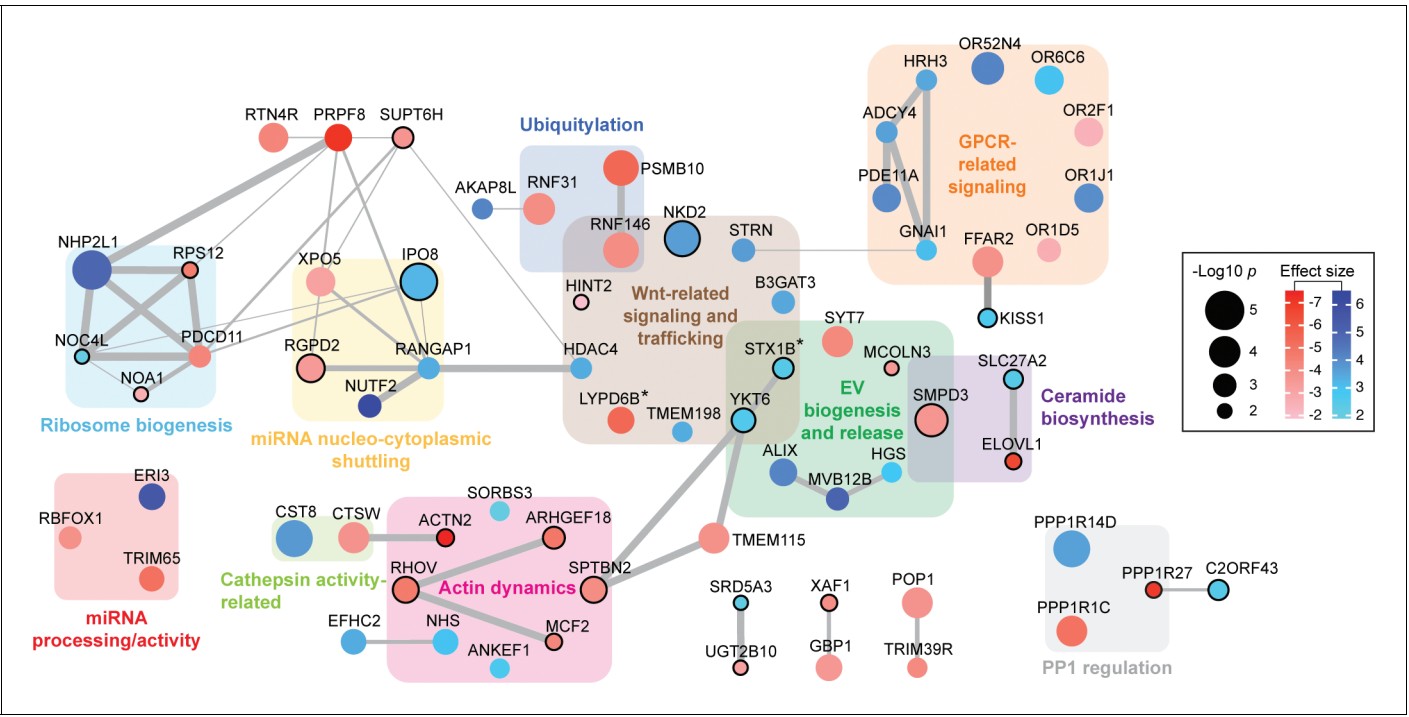

**Figure 4.** Functional network analysis of top hits. Only genes (nodes) that showed interactions in STRING analyses or that fell into a related functional category (gene clusters) are displayed. Interactions reported by STRING are represented by gray edges whose width is proportional to a combined significance score (see Materials and methods). Activators are shown in red, suppressors in blue. Node color intensity is proportional to the effect size found in the screen. Node size is proportional to significance score (-Log p-value) calculated in the screen analysis. In all panels, asterisks indicate that an isoform or close family member has been shown to play a role in the same category.

DOI: https://doi.org/10.7554/eLife.41460.007

The following figure supplement is available for figure 4:

**Figure supplement 1.** Diagram summary of roles and localizations of top screen hits.

DOI: https://doi.org/10.7554/eLife.41460.008

K562 cell lines subjected to CRISPR knockout of either SMPD3 or ARHGEF18 genes (*Figure 5A*, *Figure 5—figure supplement 1A,B*). In both cases, upon knockout of these gene products, the abundance of most miRNAs decreased ~50% in EVs but not the corresponding cellular fractions. For the miRNAs that increased significantly upon hit knockout, about half of these were seen in both replicates (*Figure 5A*). qPCR analysis of two of these endogenous miRNAs confirmed miRNA export decreases in cells depleted of SMPD3, RHOV, ARHGEF18, and BAIAP3 (*Figure 5—figure supplement 1A–D,J*). In addition, orthogonal protein assays comparing protein abundance in cellular versus EV fractions confirmed roles for SMPD3, RHOV, ARHGEF18, BAIAP3, PPP1R1C, and TREM1 in EV formation (*Figure 5*, B and C, *Figure 5—figure supplement 1A–F*). Knockout of PPP1R14D and NKD2 suppressors of bEXOmiR release yielded somewhat more variable results between experiments (*Figure 5*, B and C, *Figure 5—figure supplement 1G,H*).

Two different regulatory subunits of protein-phosphatase 1 (PP1), PPP1R1C and PPP1R14D, were detected as either activator or suppressor hits in our screen; both of these phenotypes were validated by immunoblot analysis of knockout cell lines (*Figure 5C*, *Figure 5—figure supplement 1E, H*). PP1 has been shown to play roles in yeast vacuolar fusion events as well as in exocytosis in mammalian cells (*Conradt et al., 1994*; *Peters et al., 1999*; *Sim et al., 2003*; *Gao et al., 2012*), consistent with a link to EV biogenesis regulation; moreover, the regulatory subunits are important for substrate selectivity and thus may reflect complementary phosphatase reactions.

## New players in MVE exocytosis

In an attempt to determine more precisely, the pathway(s) impacted by several of the key hits identified, we made use of a TIRF microscopy-based live cell assay to monitor MVE exocytosis

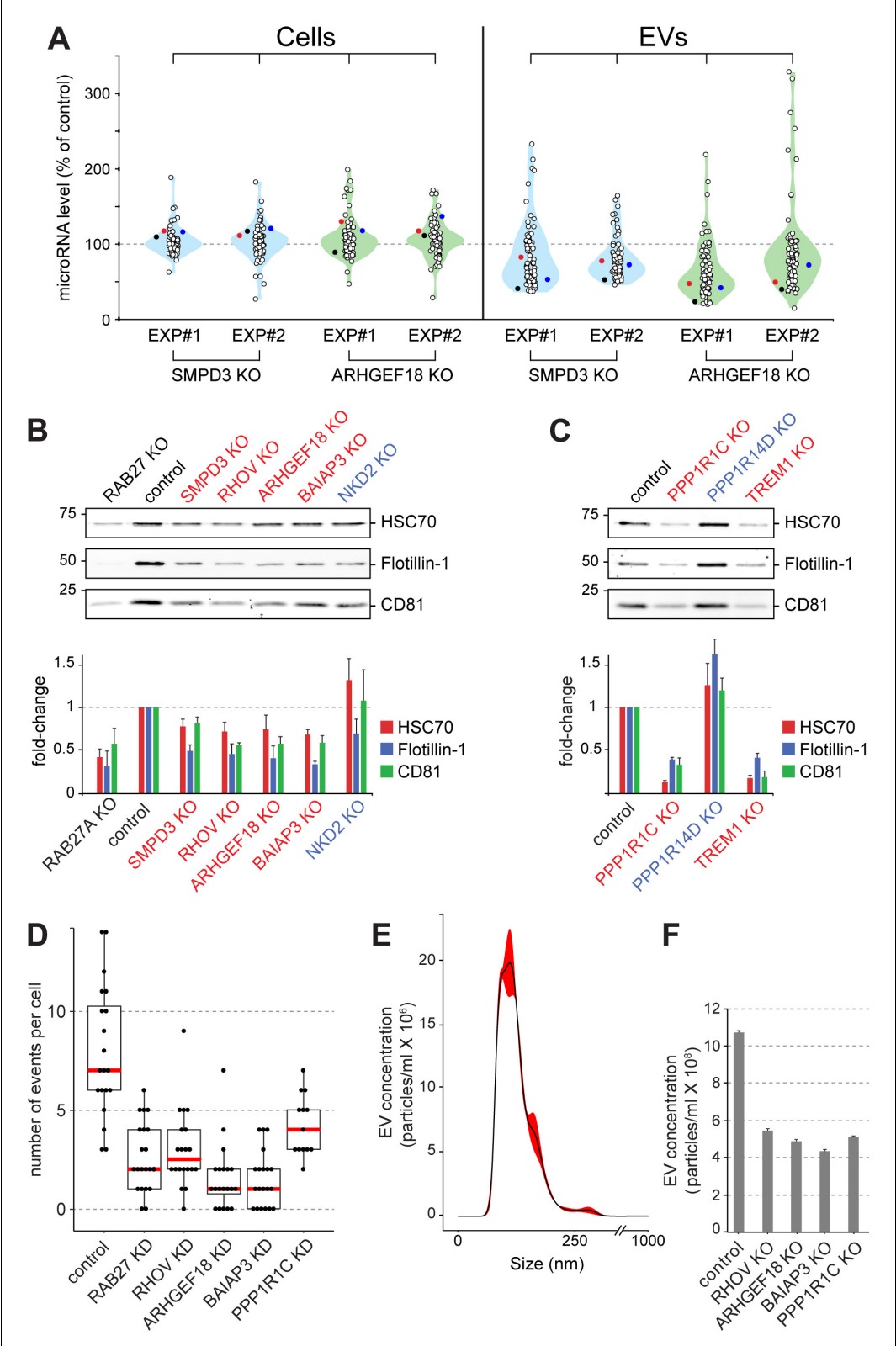

**Figure 5.** Validation of selected hits with orthogonal assays. (**A**), Quantitative Nanostring miRNA profiling of both cellular (Cells) and EV fractions was performed to compare endogenous miRNA signatures in SMPD3 CRISPR knockout (KO) or ARHGEF18 KO K562 cell lines versus control cells; blue and green violin plots, respectively. miRNA abundance is plotted as percent of miRNA abundance in control cells. Violin plots show overall distribution of common EV-associated miRNAs (circles) detected in two independent experiments. Three specific endogenous miRNAs, hsa-miR-19a-3p (black), hsa-

*Figure 5 continued on next page*

*Figure 5 continued*

miR-151a-3p (red) and hsa-miR-548b-3p (blue) are indicated. (**B**) and (**C**), Biochemical analysis of EV fractions in CRISPR knockout cell lines from selected hits derived from the initial MMM-focused (**B**) or from the rest of the genome-wide (**C**) screen. EV extracts were immunoblotted with anti-HSC70, anti-Flotillin or anti-CD81 antibodies; numbers at left indicate mobility of molecular weight markers in kD. Bars indicate standard deviation from two independent sets of two replicate experiments. (**D**) Fusion of pHluorin-CD63-positive structures at the surface of HeLa cells observed using TIRF microscopy of fluorescent siRNA-transfected cells. Each dot represents number of events per cell over 90 s. (**E**) and (**F**), Characterization of purified EVs by Nanoparticle Tracking Analysis (NTA). (**E**) Representative plot of EV size distribution. Black line and red-shaded areas indicate mean value and SEM of particle concentration at each diameter, respectively, determined after three measurements. (**F**) Yield comparison of EVs derived from control and CRISPR knockout (KO) cells.

DOI: https://doi.org/10.7554/eLife.41460.009

The following figure supplements are available for figure 5:

**Figure supplement 1.** (A–H) Confirmation of CRISPR-deletions in selected hit cell lines by TIDE analysis as indicated.
DOI: https://doi.org/10.7554/eLife.41460.010
**Figure supplement 2.** Batch retest mini screen using different sgRNA-bEXOmiR associations.
DOI: https://doi.org/10.7554/eLife.41460.011
**Figure supplement 3.** Morphology of CD63-positive compartments in HeLa cells treated with siRNAs targeting indicated genes.
DOI: https://doi.org/10.7554/eLife.41460.012

specifically, in real time, visualized using a stably expressed, pHluorin-based CD63 reporter (*Verweij et al., 2018*) in HeLa cells transfected with fluorescent siRNAs (*Video 1*). In this assay. CD63 fluorescence is only detected upon delivery to the cell surface where the protein encounters a neutral pH. As expected, RAB27A-depleted cells showed ~50% reduction in MVE exocytosis, recorded by TIRF microscopy (*Figure 5D*). In agreement with our findings (*Figure 5A–C*), siRNA-depletion of RHOV, ARHGEF18, BAIAP3, or PPP1R1C also decreased MVE exocytosis (*Figure 5D*; *Figure 5—figure supplement 1, K,L*; *Videos 1* and *2*). Thus, these genes appear to regulate CD63-positive MVE exocytosis. Nanoparticle tracking analysis (NTA) of isolated EV fractions from corresponding CRISPR knock-out cell lines further confirmed a role for these genes in the release of EV particles (*Figure 5E,F*), the size of which peaked between 100–150 nm (*Figure 5E*). Our isolation method yielded a narrow distribution of particle sizes, and only the absolute number of particles changed upon knock out of the indicated genes (*Figure 5F*). Moreover, there was no visually apparent change in the overall abundance or size of CD63-positive compartments in the cells analyzed, except for cells depleted of BAIAP3, which

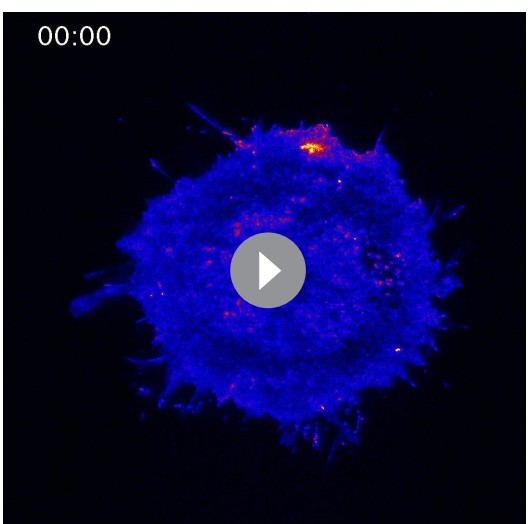

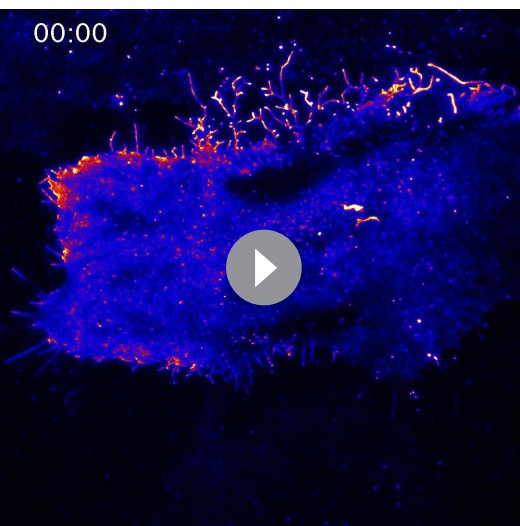

**Video 1.** Time-lapse of a control HeLa cell stably expressing pHluorin-CD63 at 3.3X real time speed shown as a heat map. Images were acquired at three frames per sec.
DOI: https://doi.org/10.7554/eLife.41460.013

**Video 2.** Time-lapse of an ARHGEF18 KD HeLa cell stably expressing pHluorin-CD63 at 3.3X real time speed as in *Video 1*.
DOI: https://doi.org/10.7554/eLife.41460.014

showed an accumulation of CD63-containing structures in the vicinity of filopodia-like structures (*Figure 5—figure supplement 3*). Together, these data suggest strongly that a block in MVE exocytosis explains the decrease in EV numbers detected for cells depleted of RHOV, ARHGEF18, BAIAP3, and PPP1R1C.

## Wnt signaling in EV biology

Wnts are growth stimulatory factors and critical developmental regulators that are also implicated in a variety of cancers (*Nusse and Clevers, 2017*). Wnts are palmitoylated in the secretory pathway and may themselves be carried to neighboring cell targets on the surface of EVs (*Nusse and Clevers, 2017*). Canonical Wnt signaling requires binding to the seven transmembrane spanning, Frizzled receptor and LRP co-receptors, to trigger a pathway that leads to the translocation of β-Catenin to the nucleus where it serves as a transcriptional co-activator (*Nusse and Clevers, 2017*). A number of Wnt-signaling modulators appeared among the hits as EV biogenesis regulators (*Figure 4*). These include LYPD6B, a relative of LYPD6 that interacts with Frizzled and LRP6 and enhances Wnt signaling (*Özhan et al., 2013*), as well as the ubiquitin ligase RNF146 that activates Wnt signaling by ubiquitinating Axins (*Zhang et al., 2011*). Knockout of these components decreased EV recovery, suggesting that Wnt signaling enhances EV release under the conditions of cell growth used in our screen. This conclusion is consistent with our discovery that loss of NKD2, a negative regulator of Wnt signaling, also enhanced bEXOmiR release in the screen (*Figure 4*; *Figure 5B*).

DeRobertis and colleagues showed previously that Wnt signaling involves intralumenal targeting of the Wnt receptor complex and associated Glycogen Synthase Kinase 3 (GSK3) and Axin into multivesicular endosomes (MVEs), in a process that requires HRS and VPS4 (*37*); this process inhibited GSK3 to enable stabilization of β-Catenin. Thus, components needed for EV release may also regulate invagination of GSK3 and Axin into MVEs. Indeed, GSK3 regulates endolysosome biogenesis via MITF, TFEB and TFE3 transcription factors (*Ploper et al., 2015*).

Wnt stimulation was shown previously to activate ARHGEF18 (*Tsuji et al., 2010*) and to up-regulate RHOV expression (*Faure and Fort, 2011*), both validated activator hits in this screen. Other genes identified here include two regulatory subunits of PP1, which showed opposite roles in EV release regulation. PP1 is a downstream modulator in both GPCR and Wnt signaling pathways. EV release is controlled by PKC-mediated SNAP23 phosphorylation, the closest homologue of SNAP25 (*Arora et al., 2017*), which mediates exocytosis upon PP1-mediated hydrolysis of PKC-phosphorylated residues (*Gao et al., 2012*).

To explore further the role of canonical Wnt signaling in EV generation, the effect of Wnt3a addition on EV release was tested in K562 cells (*Figure 6A*). Paradoxically, addition of Wnt3a to overnight serum starved cells decreased the appearance of EV markers LAMP1, HSC70 and CD81, all of which were rescued upon addition of DKK1 protein that inhibits Wnt signaling by binding to the Wnt co-receptor, LRP6 (*34*). Consistent with these findings, the GSK3 inhibitor, CHIR99021 that activates the Wnt pathway led to a significant decrease in EV levels from K562 and HEK293T cells (*Figure 6C*, *Figure 6—figure supplement 1A*). Finally, LiCl was also employed as a less specific activator of Wnt signaling and also led to a decrease in EV generation (*Figure 6—figure supplement 1B,C*).

Inhibition of GSK3 by CHIR99021 increased LAMP1 protein levels as determined by immunoblot (*Figure 6E*) and flow cytometry (*Figure 6F*), and was accompanied by an increase in lysosome numbers (*Figure 6F*); this would be expected, as Wnt-mediated GSK3 inhibition stabilizes the key endolysosomal transcriptional regulators, MITF and TFE transcription factors (*Ploper et al., 2015*; *Ploper and De Robertis, 2015*) that also regulate MVE exocytosis (*Medina et al., 2011*). Accordingly, mRNA levels of the endo-lysosomal genes, LAMP1 and MCOLN1 increased upon CHIR99021 treatment (*Figure 7A*). As expected, a robust, concomitant upregulation of AXIN2 was also observed.

Previous studies have shown that in melanosome-producing cells, RAB27A gene expression is positively regulated by MITF/TFE transcription factors (*Chiaverini et al., 2008*). According to this scenario, stabilization of these transcriptional regulators upon GSK3 inhibition would be expected to upregulate RAB27 (*Ploper and De Robertis, 2015*). Surprisingly, we observed ~50% reduction of RAB27A mRNA levels upon GSK3 inhibition in K562 cells under the conditions employed (*Figure 7A*); this was accompanied by a corresponding decrease in RAB27A protein levels (*Figure 7B*). These data likely explain the reduction of EV release observed upon CHIR99021

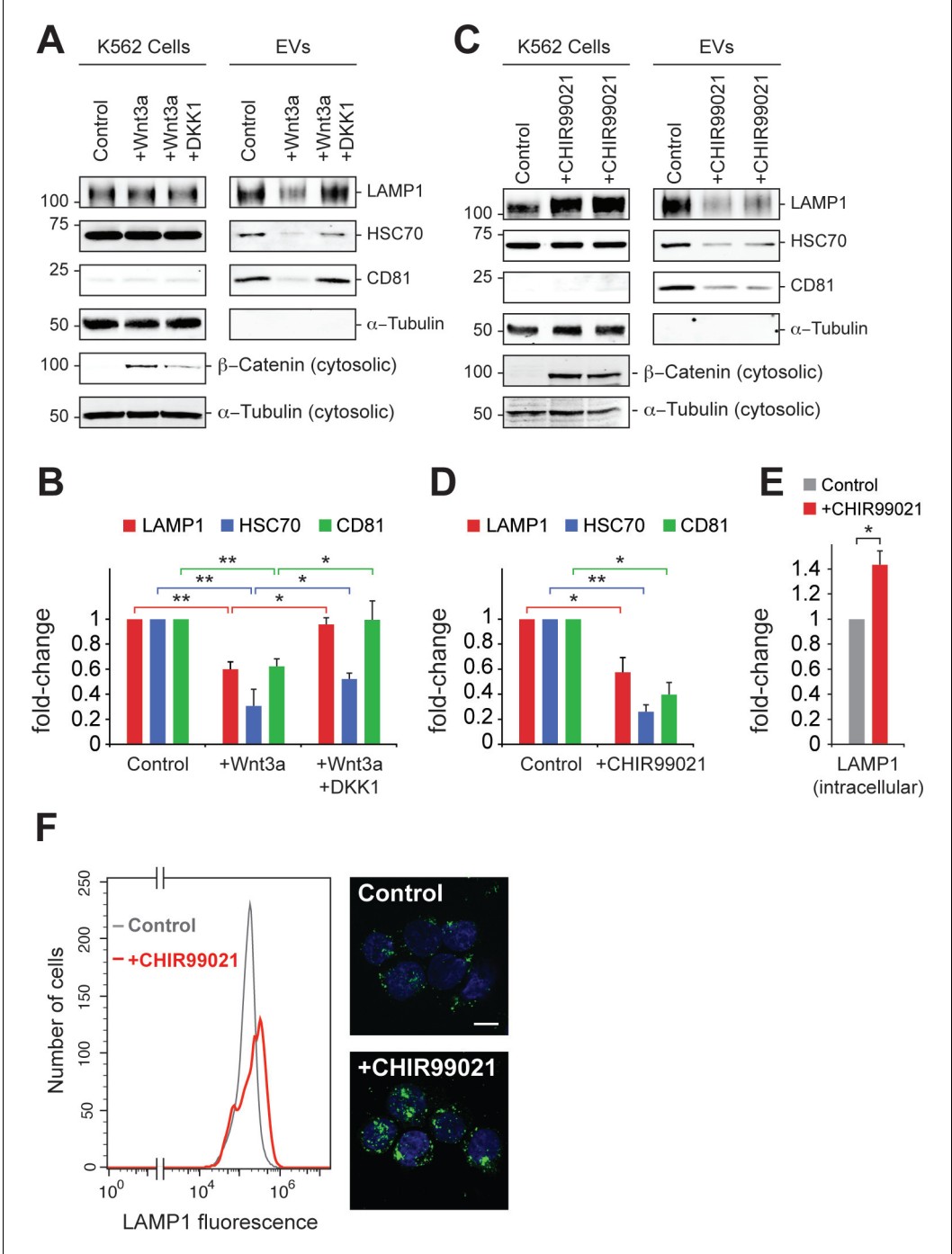

**Figure 6.** Wnt signaling regulates EV release. (**A**) Serum-starved K562 cells were treated with PBS (control) or Wnt3a (100 ng/ml) or Wnt3a + Dickkopf peptide (DKK1, 200 ng/ml) for 24 hr prior to EV isolation. Biochemical analysis of cellular (Cells) and EV fractions was performed by immunoblotting of extracts using antibodies against antigens shown. Detection of ß-Catenin was carried out using cytosolic fractions. (**B**) Quantitation of EV marker protein levels from EV immunoblots shown in (**A**). (**C**), Serum-starved K562 cells were treated with DMSO (control) or CHIR99021 (10 µM) for 24 hr prior to EV isolation. Biochemical analysis of EV fractions was performed as in (**A**). Representative immunoblots for a control experiment along with two replicate CHIR99021-treated samples are shown. (**D**) Quantitation of protein levels from EV immunoblots in (**C**). (**E**) Quantitation of LAMP1 protein from cellular fractions in (**C**). Molecular mass marker mobility is shown at the left of immunoblot panels in kilodaltons. t test: *p<0.05; **p<0.01; error bars represent SEM; $n \geq 3$. (**F**) Flow cytometry determination of LAMP1 levels in anti-

*Figure 6 continued on next page*

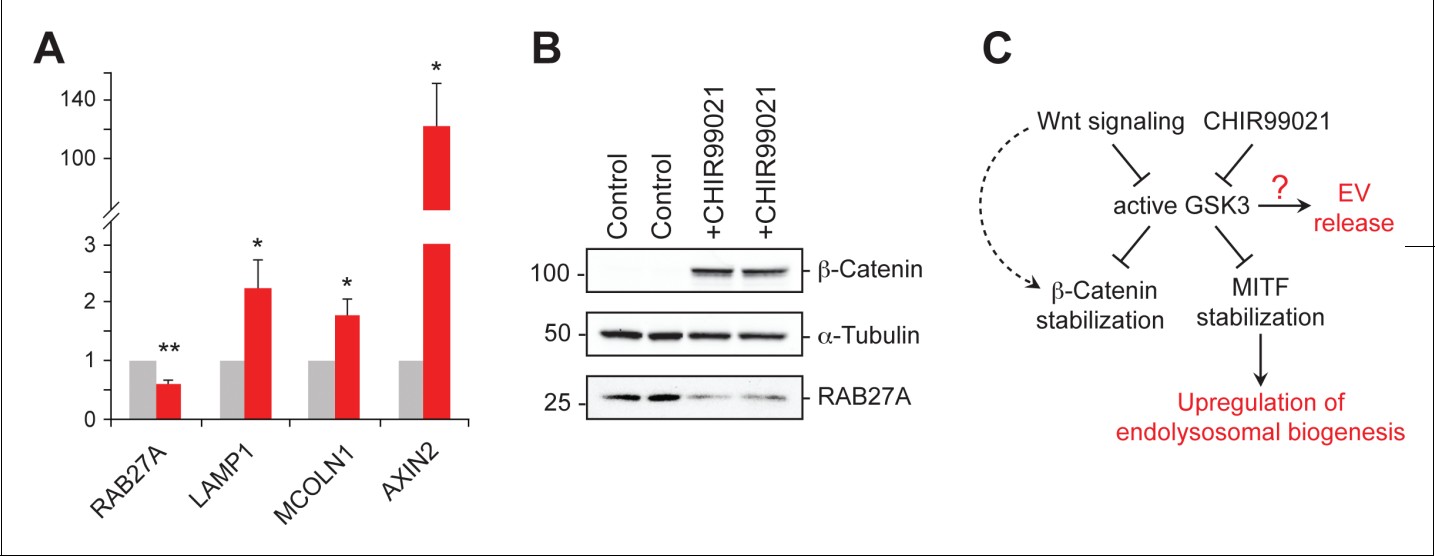

**Figure 7.** GSK3 inhibition decreases Rab27 mRNA levels. (**A**) qPCR analysis of indicated mRNA levels in K562 cells treated for 20 hr with CHIR99021. Shown is the combined data from four independent experiments; error bars represent SEM (*t* test: *p<0.05; **p<0.01). (**B**) Immunoblot of RAB27 levels in duplicate samples treated with CHIR99021 as in A. (**C**) Schematic representation of the pathway by which Wnt regulates exosome release. The link between RAB27 and GSK3 is not yet known.

DOI: https://doi.org/10.7554/eLife.41460.017

treatment, and suggest that in K562 cells, GSK3 influences RAB27 transcription in an unexpected way (*Figure 7C*).

It is important to note that in our hands, melanosome-producing RPE cells behaved differently than K562 cells upon CHIR99021 treatment: they displayed increased EV production and did not show a decrease in RAB27A protein (*Figure 6—figure supplement 1D*). Altogether, these findings indicate that cell type-specific effects (*Hooper et al., 2012*) contribute to differences in GSK3-mediated EV generation and highlight the complexity of Wnt-mediated signaling and GSK3 regulation in cells.

Why did our screen indicate that Wnt is an activator rather than a suppressor of EV release? The validation experiments were done under starvation conditions to maximize the signal—this will have additional impacts on endo-lysosomal gene regulation by TFEB and MITF regulators. How RAB27 gene expression is regulated upon GSK3 inhibition will be interesting to investigate in the future.

Although K562 cells have been reported to express both Wnts and Frizzled receptors (*Sercan et al., 2010*), we did not see a significant change in EV levels upon blocking Wnt secretion with the Wnt palmitoylation inhibitor (C59) in our parental cell cultures. This suggests that under the conditions of the screen (but not the conditions of the validation testing), knockout of various components reflected protein roles that are independent of Wnt signaling processes. Most importantly, when we tested EV release in wild type parental cells, we detected a very strong role for the GSK3 kinase in EV release regulation. Indeed, Bafilomycin A1 treatment, that activates EV release, actually inhibits Wnt signaling (*Cruciat et al., 2010*). It is very satisfying that the very EV pathway that may generate functional Wnt signals (*Nusse and Clevers, 2017*) is in fact directly regulated by Wnt signaling via an important and yet to be characterized feedback mechanism. Altogether, complex crosstalk between signaling pathways may influence EV secretion in ways that will be important to define in future work.

## Conclusion

Despite recent advances in EV research, how EV generation is regulated at the molecular level is poorly understood. We have used artificially barcoded miRNAs as surrogate reporters of EV release to enable genome-wide dissection of EV biology. We validated a number of previously reported players and discovered many new ones, including a gene cluster devoted to cortical actin dynamics and a role for ARHGEF18 in MVE exocytosis of EV-associated miRNAs. Wnt signaling- and

trafficking-related genes were also prominently represented in the functional network analysis; we found that Wnt-mediated inactivation of GSK3 tightly regulates EV release by down-regulating RAB27 mRNA and protein levels. Collectively, EV biogenesis and release appear to be part of a complex feedback mechanism that regulates Wnt signaling in cells. Finally, our screen also uncovered genes involved in intracellular trafficking and processing of miRNAs. Indeed, bEXOmiR technology can also be leveraged to study miRNA biogenesis, processing and binding to RNA-binding proteins. This genome-wide screen has provided significant leads to aid in our understanding of the molecular basis of EV generation and regulation in cells and tissues.

# Materials and methods

## Key resources table

| Reagent type (species) or resource | Designation | Source or reference | Identifiers | Additional information |
|---|---|---|---|---|
| Cell line (*Homo sapiens*) | K562 | ATCC | RRID: CVCL_0004 | |
| Cell line (*Homo sapiens*) | K562 Cas9+ | PMID: 27018887 | | |
| Cell line (*Homo sapiens*) | K562 ARHGEF18 KO #5 | this paper | | Progenitor: K562 Cas9+ |
| Cell line (*Homo sapiens*) | K562 ARHGEF18 KO #7 | this paper | | Progenitor: K562 Cas9+ |
| Cell line (*Homo sapiens*) | K562 BAIAP3 KO #8 | this paper | | Progenitor: K562 Cas9+ |
| Cell line (*Homo sapiens*) | K562 NKD2 KO #3 | this paper | | Progenitor: K562 Cas9+ |
| Cell line (*Homo sapiens*) | K562 PPP1R1C KO #4 | this paper | | Progenitor: K562 Cas9+ |
| Cell line (*Homo sapiens*) | K562 PPP1R14D KO #1 | this paper | | Progenitor: K562 Cas9+ |
| Cell line (*Homo sapiens*) | K562 RAB27A KO #1 | this paper | | Progenitor: K562 Cas9+ |
| Cell line (*Homo sapiens*) | K562 RHOV KO #10 | this paper | | Progenitor: K562 Cas9+ |
| Cell line (*Homo sapiens*) | K562 SMPD3 KO #9 | this paper | | Progenitor: K562 Cas9+ |
| Cell line (*Homo sapiens*) | K562 TREM1 KO #9 | this paper | | Progenitor: K562 Cas9+ |
| Cell line (*Homo sapiens*) | HEK293T | https://mrcppu reagents.dundee.ac.uk/ | RRID: CVCL_0045 | |
| Cell line (*Homo sapiens*) | HeLa | ATCC | RRID: CVCL_0030 | |
| Antibody | mouse monoclonal anti-a-Tubulin | Sigma-Aldrich | T5168 | (1:10000) |
| Antibody | rabbit polyclonal anti-ARHGEF18 | GeneTex | GTX102223; RRID: AB_10728594 | (1:1000) |
| Antibody | rabbit polyclonal anti-Rab27A | Synaptic Systems | 168 013 | (1:1000) |
| Antibody | mouse monoclonal anti-CD81 | BD Biosciences | 555675 | (1:1000) |
| Antibody | mouse monoclonal anti-ß-Catenin | BD Biosciences | 562505; RRID: AB_11154224 | (1:1000) |
| Antibody | mouse monoclonal anti-Golgin 97 | Thermo Scientific | A-21270 | (1:1000) |
| Antibody | rabbit polyclonal anti-Flotillin-1 | Novus Biologicals | NBP1-79022; RRID: AB_11011337 | (1:1000) |
| Antibody | mouse polyclonal anti-Calnexin | Santa Cruz | sc-23954 | (1:1000) |
| Antibody | mouse monoclonal anti-HSC70 | Santa Cruz | sc-7298 | (1:1000) |
| Antibody | mouse monoclonal anti-LAMP1 | DSHB | G1/139/5; RRID: AB_10659721 | (1:2) |
| Antibody | mouse monoclonal anti-CD63 | DSHB | H5C6; RRID: AB_528158 | (1:1000) |

*Continued on next page*

*Continued*

| Reagent type (species) or resource | Designation | Source or reference | Identifiers | Additional information |
|---|---|---|---|---|
| Antibody | rabbit polyclonal anti-GDIß | PMID: 8195183 | home-made | (1:1000) |
| Antibody | mouse monoclonal anti-LBPA | EMD Millipore | MABT837 | (1:1000) |
| Genetic reagent (*Homo sapiens*) | ARHGEF18 (siRNA) | Dharmacon | | |
| Genetic reagent (*Homo sapiens*) | BAIAP3 (siRNA) | Dharmacon | | |
| Genetic reagent (*Homo sapiens*) | RHOV (siRNA) | Dharmacon | | |
| Genetic reagent (*Homo sapiens*) | PPP1R1C (siRNA) | Dharmacon | | |
| Genetic reagent (*Homo sapiens*) | RAB27A (siRNA) | Dharmacon | | |
| Recombinant DNA reagent | pMCB306 | PMID: 28319085 | | |
| Recombinant DNA reagent | pMCB306-pHluorin-CD63 | this paper | | Progenitor: pMCB306 |
| Chemical compound, drug | Recombinant Mouse Wnt-3a Protein | R and D Sysytems | 1324-WN-002 | |
| Chemical compound, drug | CHIR-99021 | Selleckchem | S1263 | |
| Chemical compound, drug | DharmaFECT 1 | Dharmacon | | |
| Chemical compound, drug | Bafilomycin-A1 | Sigma-Aldrich | | |
| Software, algorithm | Fiji | | RRID: SCR_002285 | |
| Software, algorithm | CellProfiler | PMID: 17076895 | RRID: SCR_007358 | |
| Other | DAPI | Invitrogen | RRID: AB_2307445 | 1:10000 |

Cell culture, antibodies and immunofluorescence–HEK293T cells were grown in Dulbecco's modified Eagle's media (DMEM) containing 10% fetal bovine serum, 2 mM L-glutamine, and penicillin (100 U/ml)/streptomycin (100 µg/ml). Wild type (WT) and Cas9-expressing (*Morgens et al., 2017*) K562 cells were grown in RPMI 1640 Medium supplemented with 10% fetal bovine serum, 2 mM L-glutamine, 1 mM sodium pyruvate and penicillin (100 U/ml)/streptomycin (100 µg/ml). All cell lines were cultured at 37°C with 5% $CO_2$; cells were routinely tested for Mycoplasma using either a MycoAlert Mycoplasma Detection kit (Lonza LT07-318) or PCR. Their identity was assumed to be determined by ATCC. Extracellular vesicle-free media was prepared by overnight ultracentrifugation of RPMI supplemented with 20% FBS at 100,000 X g in a 45Ti rotor, as described (*Théry et al., 2006*).

Transfections of HEK293T cells plated in 10 cm dishes were carried out using 1.5 µg of bEXOmiR-encoding plasmid DNA (pEGFP-bEXOmiR) mixed with 1 ml serum-free DMEM and 30 µL polyethylenimine (PEI, 1 mg/ml, pH 7.0, PolySciences Inc.) and incubated at room temperature for 30 min. The mixture was added to cells that were grown for 48 hr before EV isolation. For Bafilomycin-A1 (Sigma-Aldrich) treatment, the drug was added to cell media at 10 nM and cells were cultured for 20 hr before EV collection.

Mouse monoclonal anti-α-Tubulin was from Sigma-Aldrich, rabbit polyclonal anti-ARHGEF18 was from GeneTex and anti-Rab27A was from Synaptic Systems. Mouse monoclonal anti-CD81, anti-β-Catenin and anti-Golgin 97 were from BD biosciences. Polyclonal anti-GDIβ was previously described (*Dirac-Svejstrup et al., 1994*). Rabbit polyclonal anti-Flotillin-1 antibody was from Novus Biologicals (NBP1-79022). Polyclonal anti-Calnexin and monoclonal anti-HSC70 antibodies were from Santa Cruz. Mouse monoclonal anti-LAMP1 antibody culture supernatant was from Developmental Studies

Hybridoma Bank (University of Iowa, Iowa city, IA). Mouse monoclonal anti-LBPA and anti-CD63 antibodies were from EMD Millipore (MABT837) and DHSB (H5C6), respectively.

Immunofluorescence– HeLa cells were transfected with fluorescently labeled siRNAs (DY547 label; Dharmacon) using DharmaFECT one transfection reagent, cultured for 48 hr, and replated on glass coverslips. To visualize K562 cells, cells were attached to glass coverslips using cytospin (Shandon) at 800 rpm for 5 min. Cells were then fixed with 3.7% (v/v) paraformaldehyde for 15 min, washed and permeabilized/blocked with 0.1% saponin/1% BSA-PBS before staining with anti-LAMP1 antibody followed by goat anti-rabbit Alexa 488 (1:2,000). Nuclei were stained using 0.1 µg/ml DAPI (Sigma-Aldrich). Coverslips were mounted on glass slides with Mowiol and imaged using a spinning disk confocal microscope (Yokogawa) with an electron multiplying charge-coupled device (EMCCD) camera (Andor, UK) and a 100 $\times$ 1.4 NA oil immersion objective. Finally, unbiased quantification of CD63-positive vesicles was performed using CellProfiler software (*Carpenter et al., 2006*).

Immunoblotting–HEK293T or K562 cells were harvested after 24–48 hr and lysed in lysis buffer (50 mM Hepes, pH 7.4, 150 mM NaCl, 5 mM MgCl2, and 1% Triton X-100) supplemented with protease inhibitors. After centrifugation at 12,000 g for 10 min, protein concentrations were measured in the cleared lysates. Equal amounts were resolved by SDS-PAGE followed by immunoblot analysis. EV final pellets were resuspended in SDS-PAGE loading dye (50 mM Tris-HCl pH 6.8, 6% glycerol, 2% SDS, 0.2% bromophenol blue), heated briefly at 95°C and resolved by 10% SDS-PAGE. After transfer to nitrocellulose and antibody incubation, blots were detected with ECL western blotting detection substrate or visualized using LI-COR Odyssey Imaging System. Isolation of cytosolic fractions from cell post-nuclear supernatants was performed as described (*Lu and Pfeffer, 2013*). For optimal detection of EV fraction markers (such as CD81), no reducing agent was added to the SDS-PAGE loading dye. Immunoblot signals visualized using LI-COR were analysed using ImageJ and presented as average ±SEM.

EV isolation–K562 suspension cell cultures were seeded at 5 $\times$ 10$^5$ cells/ml in EV-free RPMI and grown for 24–48 hr. Cell cultures were centrifuged at 300 $\times$ g, 5 min to pellet cells, and supernatants were centrifuged again at 2100 $\times$ g, 20 min to pellet dead cells. The resulting supernatant was spun again at 10,000 $\times$ g, 30 min to pellet cell debris and large apoptotic bodies; for large scale screen cultures this step was replaced by filtration through 0.45 µm filter units (Thermo Scientific). After this, filtration through 0.22 µm filter units (Thermo Scientific) was performed. Filtered media were ultracentrifuged for 70 min at 100,000 $\times$ g in 45Ti rotor to pellet extracellular vesicles. The supernatant was discarded and the pellet re-suspended in phosphate-buffered saline (PBS) and centrifuged again for 70 min at 100,000 $\times$ *g* in TLA100.2 rotor. Discontinuous sucrose gradient flotation was as described (*Shurtleff et al., 2016*). EV pellets were frozen in liquid nitrogen and stored at −80 °C prior to further processing.

Nanoparticle tracking analysis–EVs were purified from control and CRISPR KO K562 polyclonal cell cultures as described (see EV isolation). After final ultracentrifugation, pelleted EVs were resuspended in PBS and analyzed on a NanoSight NS300 (Malvern Panalytical) using a sCMOS camera. Three, 60 s 25 Hz movies were taken per sample. Data were analyzed using NTA 3.1 software (Malvern Panalytical).

Flow cytometry analysis of LAMP1–K562 cells were treated with either DMSO or CHIR99021. After 24 h cells were harvested and washed before fixation in 2% paraformaldehyde, permeabilization and staining with anti-LAMP1 and Alexa Fluor 488 goat anti–mouse antibodies (Invitrogen). Quantification of LAMP1 total fluorescence was performed using a BD Accuri C6 flow cytometer and data were analyzed using Flowjo software (Tree Star).

Total internal reflection fluorescence (TIRF) microscopy–HeLa cells stably expressing pHluorin-CD63 were transfected with fluorescently labeled siRNAs (DY547 label; Dharmacon) using DharmaFECT one transfection reagent, cultured for 48 hr, and replated on chambered coverglass (Thermo Scientific). After 24 hr, cells containing fluorescent siRNAs were imaged on an inverted microscope (Nikon Eclipse Ti) equipped with an electron-multiplying charge-coupled device camera (ANDOR iXon$^{EM}$+), an Apo TIRF 100X/1.49 oil DIC objective, and Nikon Perfect Focus System. Time lapse videos were acquired at 3 Hz with Nikon NIS Elements software. All imaging experiments were performed at 37°C in a humidified incubator (5% CO$_2$). Fusion events for each cell were detected and quantified as sudden increases in fluorescence above the plasma membrane background as described (*Verweij et al., 2018*). At least 20 cells were imaged per condition in two independent experiments.

bEXOmiR design–Initial development of bEXOmiRs was based on previous artificial shRNAs designs (*Chang et al., 2013*) incorporating endogenous sequences derived from the putative exosomal microRNA hsa-miR-601. Specifically, a seven nt motif (GGAGGAG, 'EXO motif') together with a contiguous loop sequence (both derived from hsa-miR-601) were placed adjacent to a 15 nt random sequence (barcode) with perfect base complementarity, positioned at the base of the miRNA stem region (*Figure 1—figure supplement 1A*). The resulting precursor hairpin was flanked by hsa-miR-30a context sequences. For the initial bEXOmiR test library, a custom C-script was written to design barcode sequences based upon specific rules: 40–60% GC content, minimum Hamming distance of 4 or more between any two barcodes, and absence of homopolymers (AAAA, UUUU, CCCC, etc.) and restriction enzyme cut sites used for cloning purposes. With this strategy, 2500 bEXOmiRs were designed, and two versions of each were used, one with an additional A-C mismatch added to ensure proper primary-miR processing (*Chang et al., 2013*) for a total of 5000 bEXOmiRs.

To generate ~25,000 bEXOmiRs used in the MMM screen sublibrary, 15 base-pair barcodes were similarly generated with a minimum Hamming distance of four, moderate GC content (40–60%), and absence of homopolyers (GGGG, AAAA etc) and restriction sites. The first base-pair was altered to create a mismatch at the base of the hairpin to ensure proper processing (either A-C, T-C, G-A, or C-A). For subsequent sublibraries, this ~25,000 bEXOmiR set was filtered for either bEXOmiRs that failed to be detected in WT exosome samples (counts less than 10) or were detected at such high representation in WT exosomes as to interfere with sequencing depth (count >10,000). Additional barcodes were randomly generated as above for larger libraries.

Stem-loop RT-PCR and qRT-PCR–For bEXOmiR and endogenous miRNA detection in cellular or EV fractions, RNA was extracted from intact HEK293T or K562 cells, or isolated EVs using TRIzol, according to the manufacturer. EV RNAs (200–500 ng) were reverse transcribed into cDNA using Stem-Loop primers and SuperScript III FirstStrand Synthesis System (Life Technologies) according to a pulsed RT protocol (*Varkonyi-Gasic and Hellens, 2011*). Stem-loop RT primers are summarized in *Supplementary file 3*. qRT-PCR experiments used a qPCR mix recipe: 1X Phusion HF buffer (NEB), 200 µM dNTPs, 1X SYBR green I (Invitrogen), HighFidelity DNA Polymerase (M0530S, NEB), and 100 nM each primer. The following programs were used: (98°C, 3', then 40X cycles of 98°C 15 s, 60°C 20 s, 72°C 3 s). A synthetic spike-in RT RNA (*Supplementary file 3*) was added to each EV sample to control for variability in the starting material and the downstream RNA isolation step. The sequences of PCR primers are summarized in *Supplementary file 3*. qPCR was run in a BioRad CFX96 Real-Time System. PCR amplification products were analyzed on 4% agarose gels stained with ethidium bromide. For qPCR experiments related to *Figure 7*, total cell RNA was reverse transcribed into cDNA using oligo(dT)20 primers and SuperScript III FirstStrand Synthesis System (Life Technologies). qPCR was performed using iTaq Universal SYBR Green Supermix (BioRad) using gene-specific primers (*Supplementary file 3*) and run in a ViiA 7 Real-Time PCR System (Thermo Fisher Scientific). Transcript levels relative to HPRT1 were calculated using the ΔCt method.

RNase protection assay–K562 cells expressing bEXOmiRs were grown for 24 hr in RPMI and EVs were isolated from conditioned media. EVs were then treated with RNase A (10 µg/ml in PBS)±Triton X-100 (0.25%), or with PBS alone, for 30 min at room temperature. RNase inhibitor (150 U/ml, Thermo Fisher Scientific) was then added and EV-RNA isolated with TRIzol reagent (Life Technologies).

Construction of sgRNA-bEXOmiR plasmid libraries–Oligonucleotides encoding sgRNAs (*Morgens et al., 2017*) coupled to individual bEXOmiRs were synthesized as pooled libraries (Agilent). BlpI and EcoRI restriction sites were introduced between sgRNA and bEXOmiR sequences. Focused sublibraries were PCR-amplified with primer sequences common to each sublibrary, digested with BstXI and MfeI enzymes, ligated into BstXI/EcoRI-digested pMCB306 vector, and transformed into electro-competent cells (Lucigen). Plasmid DNA was isolated using a QIAGEN kit. Pooled plasmids were then digested with BlpI and EcoRI, and an expression cassette encoding a puromycin-resistance gene and a GFP-ORF separated by a T2A sequence under EF-1 alpha promoter, cut with the same restriction enzyme pair, was inserted between sgRNA and bEXOmiR sequences. This generated nine focused sublibraries (*48, Supplementary file 4*), which together contained ~210,000 elements targeting ~20,000 genes with ~13,000 non-targeting and safe-targeting controls (*Morgens et al., 2017*); 10 sgRNA per gene, each sgRNA associated to a unique bEXOmiR).

Lentivirus production and cell infection–Lentivirus production and infection were performed as described (*Kampmann et al., 2014*). Briefly, HEK293T cells were co-transfected with bEXOmiR libraries or individual sgRNA vectors and third generation packaging plasmids (pVSV-G, pMDL, pRSV) using polyethylenimine. Lentivirus was harvested after 48 h-72 h and filtered through a 0.45 µm filter (Millipore). A Spinfection method was used to deliver sgRNA-bEXOmiR lentiviral plasmid sublibraries into K562 cells. Briefly, K562 cells were re-suspended in lentiviral supernatant supplemented with 8 µg/ml polybrene, distributed into 6-well plates, and centrifuged for 2 hr at 1000 × *g*, 33°C. Lentiviral supernatant was removed and cells were grown in fresh RPMI. After 48 hr recovery, 1.75 µg/ml of puromycin was added to select the cells for stable integration of the virus. Selection was monitored by flow cytometry and puromycin was removed from the media once >95% of the cells were GFP-positive.

CRISPR/Cas9-bEXOmiR screens in K562 cells–For genome-wide coverage, a total of nine, focused CRISPR/Cas9 deletion screens using nine different sublibraries of sgRNA-bEXOmiR associations (*Supplementary file 4*) were performed. WT and Cas9-expressing K562 cells were infected at low MOI (<1). Transduced cells were selected and expanded in puromycin-supplemented media over 5–7 days before conducting experiments at 8–10 days post-infection. Initially, we performed two independent pilot screens focused on membrane trafficking, mitochondrial and motility genes (MMM screen). Subsequent screens were conducted in duplicate to further minimize technical variation observed in previous screens. Screens were started at 10,000–15,000 fold sublibrary coverage (elements per ml of cell culture) to ensure maximal bEXOmiR representation in EVs but also to minimize variability due to downstream loss of barcodes during RNA isolation and fractionation. Isolation of EVs was performed 48 hr later using protocol P2 (*Figure 1—figure supplement 1B*). Cellular fractions with at least 1000-fold sublibrary coverage were collected at the start (0 hr) and end (48 hr) of each screen.

bEXOmiR isolation and purification from screen EV RNA–Total RNA was extracted from each EV sample using TRIzol reagent (Life Technologies), as described (*Lässer, 2013*). The resulting RNA pellet was re-suspended in nuclease-free water and equal amount of 2x RNA loading dye was added (90% formamide, 50 mM HEPES, bromophenol blue, xylene cyanide). Fractionation of total RNA was carried out on denaturing polyacrylamide gels containing 7% urea/15% acrylamide in Tris-borate-EDTA buffer (TBE). Custom 21 and 24 bp single-stranded RNA markers were used to indicate approximate miRNA migration on a gel from which gel band excision was monitored by UV shadowing. miRNAs were then gel-purified, reconstituted in nuclease-free water, and stored in −80 °C.

Small RNA library preparation and deep sequencing–bEXOmiR sequencing libraries from isolated EV miRNA was prepared using a modified protocol from TruSeq Small RNA Library Preparation Kit (Illumina). For each library,~1 µg of fractionated small RNA was ligated first to a 3'-adapter and then to 5'-adapter, followed by reverse transcription with custom primer (*Supplementary file 3*) containing a fragment complementary to the EXO motif in bEXOmiRs (to specifically enrich for bEXOmiRs in the final sequencing library). Resulting cDNA was then uniquely indexed by PCR to generate the final sequencing library. Initial sequencing trials showed some rRNA contaminant fragments (specially 45S5) that were predominant in our sequencing data, which compromised sequencing depth. To remove rRNA 45S5, a specific hairpin oligo blocker (*Supplementary file 3*) containing a sequence complementary to the predominant rRNA fragment was incubated after ligation of the 3' adapter to the miRNA (*Roberts et al., 2015*). This oligo blocked subsequent ligation of the 5' adapter to the contaminant rRNA fragment, thus excluding it from later steps. After PCR amplification, indexed samples were separated on a 4.5% low melting point agarose gel and products corresponding to 140 bp were purified using a MinElute PCR Purification Kit (Qiagen). RNA concentration was determined by Qubit fluorometer (Thermo Fisher Scientific). Pooled libraries were denatured with NaOH and diluted to 2.3 pM according to the Illumina protocol, and sequenced using a NextSeq 500/550 High Output v2 kit on NextSeq 500 system (Illumina).

Sequencing data analysis–Sequencing of sgRNAs from DNA was trimmed and aligned using Bowtie version 1.1.2 with zero mismatches tolerated (*Morgens et al., 2016*). All counts from multi-mapping reads were used. For sequencing of barcodes from RNA, reads were trimmed to 14 base-pairs and aligned to the libraries using Bowtie version 1.1.2 with two mismatches tolerated and all multi-mapping reads used. *Supplementary file 5* and *6* contain bEXOmiR and sgRNA sequencing counts derived from each duplicated screen. Gene-level effects and an associated log-likelihood ratio (confidence score) were then calculated with casTLE as previously described (*Morgens et al., 2016*). For

each gene, casTLE combines measurements from 10 different bEXOmiRs (associated with the sgRNAs targeting each gene) and gives an effect size estimate for the gene perturbation (gene K. O.) as well as a p-value associated with that effect. casTLE estimates the maximum possible phenotype such that targeting elements are most likely to be found between this phenotype and zero. Additionally, casTLE allows data from multiple screen types or from replicates of the same screen type to be compared and combined by finding a single effect size consistent with all data (*Morgens et al., 2016*). Briefly, low count (<10) bEXOmiRs were filtered, and a median-normalized log count ratio was calculated for each barcode. Distributions of barcode enrichments corresponding to each gene were compared to the distribution of nontargeting and safe-targeting sgRNAs. P-values were then calculated by permuting bEXOmiRs corresponding to targeting sgRNAs. Code is available at https://bitbucket.org/dmorgens/castle (Copy archived at https://github.com/elifescien-ces-publications/dmorgens-castle).

Nanostring profiling was performed with the nCounter Human v3 microRNA expression assay. Control, SMPD3 and ARHGEF18 CRISPR-KO K562 polyclonal cell lines were seeded at $2.5 \times 10^5$ cells/ml in EV-free RPMI medium and grown for 48 hr before EV isolation. RNA from cellular and exosomal fractions was extracted using Trizol as described (*Lässer, 2013*). EV RNA was resuspended in 4 μL $H_2O$ and 3 μL (~300 ng) from each sample were used in the assay; 200 ng cellular RNA was used. RNA integrity was confirmed by Bioanalyzer before conducting the assay. Around 20 amol of synthetic spike-in RNA oligo (ath-miR-159a, see *Supplementary file 3*) was added to each EV sample prior to RNA extraction and used as a normalization factor in downstream EV miRNA data analysis. Raw read data from the nCounter microRNA assay were analyzed as described in the nCounter Expression Data Analysis Guide (NanoString Technologies). Data normalization for cellular miRs was perfomed using the geometric mean of the top 100 expressed miRNAs. Only miRs that passed signal-to-noise cut-off values and that were consistently detected in either cellular or EV fractions (274 and 74 miRNAs, respectively) throughout all samples, were included in the analysis and plotted using R software. Normalized miRNA counts (*Supplementary file 7*) from SMPD3 KO or ARHGEF18 KO samples were divided by control miRNA counts from independent experiments (n = 2) and plotted as percent of control miRNA abundance.

Generation of CRISPR-KO cell lines for validation experiments–Polyclonal, CRISPR-KO cell lines were generated by transducing K562 stably expressing SFFV-Cas9-BFP with Lentiviral vectors encoding hit-targeting sgRNAs; at least two different sgRNAs per hit were tested. Cultures were incubated for at least 7–10 d with the sgRNAs to ensure adequate depletion of target-protein levels before conducting experiments. TIDE analysis (https://tide.nki.nl/) was performed to assess CRISPR deletions. Briefly, genomic DNA was extracted from cells using a QIAamp DNA mini kit (Qiagen). A 500–700 bp region containing the sgRNA cut site was amplified by PCR using GoTaq Polymerase (Promega). PCR amplification products were separated on a 0.75% TAE-agarose gel, from which amplicon bands were excised, and DNA was purified from the gel slice using a QIAquick Gel Extraction kit (Qiagen). Sanger sequencing of PCR products followed by trace decomposition analysis with TIDE software (https://tide.nki.nl/) was used to determine indel frequency in CRISPR-KO cell lines compared with control cells expressing a control sgRNA (targeting a genomic safe-harbor region). For validation experiments, sgRNA-expressing cell lines that showed higher degree of knockout efficiency as determined by either TIDE and/or immunoblot analysis were used (*Figure 5—figure supplement 1*).

Wnt signaling–For Wnt3a stimulation, K562 cells were starved overnight in serum-free conditions. The next day, the medium was replaced with fresh EV-free RPMI supplemented with 100 ng/ml recombinant Wnt3a protein (R and D Systems), or Wnt3a combined with 200 ng/ml recombinant DKK1 protein (R and D Systems) and EVs were collected 24 hr later. The GSK3 specific inhibitor, CHIR99021 (Selleckchem) and LiCl were used at 10 μM and 10 mM, respectively for 24 hr before collection of EVs from the medium. Quantitation of immunoblots from each of these conditions was normalized to account for cell viability, as measured by flow cytometry (BD Accuri). C59 Porcupine inhibitor was the generous gift of Dr. Roel Nusse, Stanford University.

Functional Network analysis–Top hits were collectively queried in STRING (https://string-db.org/) to search for gene interactions according to co-expression, experimental or curated database sources. The resulting STRING file reporting pair-wise gene interactions was imported into Cytoscape software (*Shannon et al., 2003*) to create a visually comprehensive functional interactome. Interactions were represented as edges whose thickness was proportional to a calculated combined score

derived from STRING analysis. The final Cytoscape map was further manually curated to include additional screen hits (nodes) that despite not being reported to interact, could still be clustered in a common functional category revealed in the screen.

## Data and materials availability

All data is available in the main text or the supplementary files. All code and materials used in the analysis are available to any researcher for purposes of reproducing or extending the analysis.

## Acknowledgments

We thank Drs. Rhiju Das, Rajat Rohatgi and Roel Nusse for helpful discussions. We also thank Sam Gambhir and Jung Ho Yu for help with NTA analysis. This research was funded by the NIH (DK37332 to SRP, DP2HD084069 to MCB, T32 HG000044 to DWM).

## Additional information

### Competing interests

Suzanne R Pfeffer: Reviewing editor, *eLife*. The other authors declare that no competing interests exist.

### Funding

| Funder | Grant reference number | Author |
|---|---|---|
| National Institutes of Health | NIDDK 37332 | Suzanne R Pfeffer |
| National Institutes of Health | DP2HD084069 | Michael C Bassik |
| National Institutes of Health | T32 HG000044 | David W Morgens |

The funders had no role in study design, data collection and interpretation, or the decision to submit the work for publication.

### Author contributions

Albert Lu, Conceptualization, Data curation, Formal analysis, Validation, Investigation, Visualization, Methodology, Writing—original draft, Project administration, Writing—review and editing; Paulina Wawro, Investigation, Methodology, Writing—review and editing; David W Morgens, Data curation, Software, Formal analysis, Validation, Methodology, Writing—review and editing; Fernando Portela, Data curation, Software, Methodology, Writing—original draft, Writing—review and editing; Michael C Bassik, Conceptualization, Resources, Data curation, Software, Funding acquisition, Methodology, Writing—review and editing; Suzanne R Pfeffer, Conceptualization, Resources, Data curation, Formal analysis, Supervision, Funding acquisition, Methodology, Writing—original draft, Project administration, Writing—review and editing

### Author ORCIDs

Fernando Portela (iD) http://orcid.org/0000-0003-0238-9251
Suzanne R Pfeffer (iD) http://orcid.org/0000-0002-6462-984X

### Decision letter and Author response

Decision letter https://doi.org/10.7554/eLife.41460.027
Author response https://doi.org/10.7554/eLife.41460.028

## Additional files

### Supplementary files

• Supplementary file 1. Screen results from EV and cellular fractions.
DOI: https://doi.org/10.7554/eLife.41460.018

- Supplementary file 2. Data from all sublibrary screens with bEXOmiR recovery rates.
DOI: https://doi.org/10.7554/eLife.41460.019
- Supplementary file 3. Sequences of oligonucleotides used in this study.
DOI: https://doi.org/10.7554/eLife.41460.020
- Supplementary file 4. Sequences of sgRNA-bEXOmiR associations used in the genome-wide screen.
DOI: https://doi.org/10.7554/eLife.41460.021
- Supplementary file 5. Sequencing counts of bEXOmiRs in EV fraction sub-library screens.
DOI: https://doi.org/10.7554/eLife.41460.022
- Supplementary file 6. Sequencing counts of sgRNAs in cellular fractions from sub-library screens.
DOI: https://doi.org/10.7554/eLife.41460.023
- Supplementary file 7. NanoString miRNA profiling results.
DOI: https://doi.org/10.7554/eLife.41460.024
- Transparent reporting form
DOI: https://doi.org/10.7554/eLife.41460.025

### Data availability

All data generated or analysed during this study are included in the manuscript and supporting files.

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
