## [Decision Letter]

Thank you for sending your article entitled "Genome-wide interrogation of extracellular vesicle release using barcoded miRNAs reveals a role for Wnt signaling" for peer review at *eLife*. Your article is being evaluated by Anna Akhmanova as the Senior Editor, a Reviewing Editor, and three reviewers.

As you will see from the reviews, the reviewers' opinions of your study were mixed. All reviewers feel that the bEXOmiR experimental approach is clever and may hold promise for identifying EV biogenesis pathway components. However, while the reviewers acknowledge the potential trove of new information regarding EV biogenesis, they found the breadth and diversity of the 'hits' a matter of some concern, because for most cases it is not possible to distinguish "fundamental" versus "peripheral" hits in EV biogenesis pathways. It is a fair assumption that many of the hits impact EV biogenesis and release due to general perturbations to organelle function (that is, they do not directly participate in some aspect of EV biology). The consensus view is that additional experimentation is necessary to sufficiently validate the bEXOmiR approach and the screening data, and the reviewers request that you outline additional experimentation to address this. The reviewers would be especially encouraged by experimental data regarding mechanism: what aspect of EV biogenesis is affected by loss of ARHGEF18 and/or is regulated by Wnt signaling? The reviewers felt that the Wnt signaling and ARHGEF18 follow-up work was largely confirmatory of the original screen hits but did not provide strong additional evidence for a fundamental role of these factors in EV biogenesis or regulation. The reviewers would also have liked to see some additional data that would provide insight into the origin of the bEXOmiR-containing EV vesicle (plasma membrane? endosome? another organelle?).

Reviewer #1:

The manuscript describes the construction and validation of a genetic screening method that links specific sgRNA sequences with synthetic miRNAs that are secreted via extra-cellular vesicles (EVs). The method allows one to use deep sequencing to gauge the effects of targeted gene knockouts on EV-mediated release of miRNAs on a whole genome scale. The method was applied in replicate screens, and its effectiveness is suggested by the identification of a handful of gene products previously implicated in EV biogenesis and/or release. In addition, at least one gene, ARHGEF18, was shown to be required for EV-mediated release of endogenous miRNAs.

I find the method to be a clever implementation of miRNA and sgRNA processing pathways to link genetics with phenotype. The most important question is, does it work? The identification of gene products previously identified as required for EV release suggests that it does. The approach did, however, require the authors to tackle a few technical bugs (e.g., DNA sequencing, effects of different sgRNA-barcode combinations, etc.) and it is not clear if further, unknown biases (if any) impact the utility of the method as a general method for evaluating the genetics of EV biogenesis. A large number of genes not previously implicated in EV biology are asserted to play roles in EV biogenesis. At this stage, it is unclear if any of the 'new' hits play a direct role in EV cargo sorting, EV biogenesis, or EV release. As with most screens, the ultimate utility of the data will require substantial amount of follow-up work and will not be known for years to come.

*Reviewer #2:*

In this manuscript Lu and colleagues perform a genome wide screen to get insights into the biogenesis and regulation of Extracellular Vesicle release.

For this purpose, the authors set up an elegant system based on the generation of barcoded miRNAs that depict a sequence that has been previously shown to target miRNAs to multivesicular endosomes of T lymphocytes and in particular to their intralumenal vesicles (called exosomes once they are secreted). These findings from the group of Sanchez Madrid are largely exploited here (and would deserve better acknowledgement).

The barcoded miRNAs are then used as reporters of EV release which is followed by a genome wide screen coupled with CRISPR/Cas9.

The authors have performed an incredible amount of work, that certainly is a very interesting set up.

There are several concerns with the manuscript, interpretation of results and the claims. Finally, one is a little left with the question on how such approach can really answer fundamental questions and pinpoint general or particular/specific machineries required for EV biogenesis and release.

Essential revisions:

1) EVs comprise very different types of vesicles that are either endosome derived (so called exosomes) and plasma membrane derived. One important issue in the field is the possibility in the future to ascribe functions to the different subpopulations. Several studies have described and reported distinct machineries for the biogenesis and release of exosomes and microvesicles that can be cell type and cargo specific (summarized in different recent reviews from different authors). Therefore, it is unclear what is the approach that is here developed bringing to the complexity that has been observed and described for the very different systems.

2) It is unclear from the claims in the manuscript and from the experiments done if one is trying to get hints on the biogenesis and release of endosome-derived exosomes or plasma membrane derived vesicles. It would certainly be important for the general audience to be clearer on this issue that is a real concern.

With this in mind, the characterization made, that is far from being very extensive, would seem to indicate that the authors maybe rather focus on endosome derived exosomes.

This point should be consolidated better with additional proteins tested by Western blot, CD63, Tsg101. For example, CD63 should rather be used for the immunofluorescence experiments presented. Lamp1, if present in late endosomes, is much more enriched in lysosomes. The experiments presented in Figure 1 and Figure 6 are difficult then to interpret.

Lamp1 is far from being the best "exosome" marker as in most cells Lamp1 is on the limiting membrane of the organelle and it is Lamp2 that is rather on the intraluminal vesicles. Therefore, it would be important to control whether in the cells under study Lamp1 is targeted to the intraluminal vesicles (or even if this is a consequence of the barcoded miRNAs?)

3) Following point 2, electron microscopy combined with extra immunofluorescence imaging would be more then appropriate and required to be sure, even to set background for the all findings, that the barcoded miRNAs, once expressed are not impacting in cell/organelle morphology and/or if they are targeted to the right place. This could also help understanding the effects of Rab27A which in most cells where it is expressed (not all cells express the "a" isoform) is present in Golgi TGN. Rab27a is localized to Lysosome Related organelles only in specialized cells. Therefore, is Rab27a in these manipulated cells recruited to multivesicular endosomes that may specialize after introduction of the barcoded miRs? For example, this is the case in HeLa cells expressing MHC Class II and all associated molecules for antigen presentation (Ostrowski et al., 2010). The sequence used has been described to my knowledge only in T cells, is it really driving the miRs to the right place in very different cells? In conclusion extra controls would support part of the claims. This would then help understanding what subpopulation of EVs is targeted in the study.

4) I would be more careful generalizing the concept based on the two cell lines used in this study although both express wnt.

5) The manuscript describes the involvement of Wnt signaling in the regulation of « EV biogenesis ». For me it is unclear what is really the mechanism of Wnt action: is it on the biogenesis per se or on the release/secretion step which are different parts of the pathway? It could be only on the release but whether there are biogenetic functions remains unclear. Are vesicles still formed? Is the cargo still loaded on vesicles? One has pretty much the tendency to put together biogenesis/release but the mechanisms involved in the generation and then in the release are very distinct. The manuscript should be clear on this point.

6) In subsection “Wnt signaling in EV biology” the authors test the effect of Wnt3a on EV release and show a decrease (is it on biogenesis or the release? by which mechanism?). This appears different from what was previously described in other cells such as microglia (Hooper et al., 2012), where Wnt3a induces exosome release, and inhibition of GSK3 does not altered exosome secretion, indicating that exosome release occurs through a GSK3-independent mechanism. Such discrepancy is not described, and the paper not quoted if I am not wrong.

7) In concern to the effects of YKT6 that has been described by the group of Boutros as required for exosome-dependent Wnt release in *Drosophila* wing disc and cells and Hela cells, the paper of Gross et al., 2012 is not quoted.

8) In Figure 1—figure supplement 1B and in the legend there is a discrepancy: in the figure it is 12.200g and in the legend 10.000 g and the time of not indicated in the figure.

9) Figure 5B in the text there is a reference to cells versus EVs (subsection “A CRISPR/Cas9 screen using bEXOmiRs”) though unclear in the Figure.

In conclusion, it is an elegant set up that deserves attention but certainly additional controls. At this point in my opinion it is more the set-up of a new method that maybe will set the stage for more meaningful experiments in the future. This may help to better target potential mechanisms and therefore understand the crosstalks and pathways influencing EV (rather subpopulations of EVs) biogenesis and release which should be separated as this are very different processes.

Reviewer #3:

In this manuscript, Albert Lu and colleagues, have designed a screen to identify regulators of the secretion of extracellular vesicles (EV). They have used a clever design by combining bar-coding of the EV with a single guide RNA to target genes using the Cas9 enzyme. Bar-coding is achieved using custom-designed miRNAs, the bEXOmiR.

The authors used a pool screen format where an excess of cells is transfected with a complex library of viruses. Next generation sequencing then quantifies the abundance of specific miRNA in the supernatant of the pool of cells. Because each gene-specific guide RNA is combined with a specific bEXOmiR, inference can be drawn from the levels of a bEXOmiR in the supernatant.

This is an elegant study that provides a valuable resource for researchers interested in EV. It also provides a proof of concept that such bar-coding of EV is feasible. Finally, it demonstrates elegantly that Wnt signaling is involved in regulating EV secretion.

In such pooled screens, the normalisation steps are really key to avoid mis-interpreting the data. In this case, Lu and colleagues have normalised first the bEXOmiR count to a cell line devoid of the nuclease Cas9. This should exclude any effect that the bEXOmiR barcode could have on packaging into EV or the effect the miRNAs could have on gene expression. The authors show convincingly that this is the case.

The next concern would be that the CRISPR targeted gene would affect the synthesis of the bar-coded miRNA instead of packaging and export in EV. These are admittedly two very different processes and it would be advantageous to distinguish them.

As far as I could see, the authors did not use the comparison extra/intra cellular levels of bEXOmiR to sort their hits. It would be interesting to sort the hits based on whether there is an individual correlation between intracellular and extracellular amount of bEXOmiR upon gene knockout. This should be the case for the hits proposed to be involved in "miRNA processing and trafficking" (proposal based on previous annotations) (Figure 4). In theory it could help sort the other hits between miRNA processing and EV biosynthesis, especially for RNA binding proteins.

The final concern with screen design would be whether some genes could affect the uptake and degradation of EV. Indeed, as this is a suspension culture, cells have in theory the capacity to affect the extracellular levels of EV also through endocytosis and degradation. A hit that would affect the membrane surface of EV and thus affect EV binding to neighbouring cells after secretion could, in theory affect the overall levels of EV without affecting their synthesis and release. I wonder if this could be an explanation for the effect of some of the extracellular proteins identified (Figure 4). I wonder if the authors can exclude this effect or control for it?

" Unexpectedly, YKT6, which appeared as a suppressor in the initial screen (Figure 2B), was uncovered as a slight activator instead; this may be related to differences in the bEXOmiRs associated with this gene in the two separate analyses..… This discrepancy highlights the importance of orthogonal hit validation experiments."

It also highlights the difficulty in interpreting the significance score (-Log10(P-value) to evaluate reproducibility. I have trouble evaluating how many pairs of bEXOmiRs-sgRNA gave similar results with this metric.

It would be surprising that the YKT6 case would occur if the result found in the main screen is attributable to 3 or 4 pairs of bEXOmiR+sgRNA. I think it would help clarify the data if the authors provided a table with the result for each gene: with how many constructs gave a result and what score.

---

## [Author Response]

As you will see from the reviews, the reviewers' opinions of your study were mixed. All reviewers feel that the bEXOmiR experimental approach is clever and may hold promise for identifying EV biogenesis pathway components. However, while the reviewers acknowledge the potential trove of new information regarding EV biogenesis, they found the breadth and diversity of the 'hits' a matter of some concern, because for most cases it is not possible to distinguish "fundamental" versus "peripheral" hits in EV biogenesis pathways. It is a fair assumption that many of the hits impact EV biogenesis and release due to general perturbations to organelle function (that is, they do not directly participate in some aspect of EV biology). The consensus view is that additional experimentation is necessary to sufficiently validate the bEXOmiR approach and the screening data, and the reviewers request that you outline additional experimentation to address this. The reviewers would be especially encouraged by experimental data regarding mechanism: what aspect of EV biogenesis is affected by loss of ARHGEF18 and/or is regulated by Wnt signaling? The reviewers felt that the Wnt signaling and ARHGEF18 follow-up work was largely confirmatory of the original screen hits but did not provide strong additional evidence for a fundamental role of these factors in EV biogenesis or regulation. The reviewers would also have liked to see some additional data that would provide insight into the origin of the bEXOmiR-containing EV vesicle (plasma membrane? endosome? another organelle?).

We believe that the live cell imaging shows that loss of EVs from our studied hits is due to decreased exocytosis of multivesicular endosomes – a very important new finding. In addition, we provide a molecular explanation for how the GSK3 inhibition blocks K562 MVE release – by decreasing Rab27 mRNA and protein levels.

Reviewer #1:The manuscript describes the construction and validation of a genetic screening method that links specific sgRNA sequences with synthetic miRNAs that are secreted via extra-cellular vesicles (EVs). The method allows one to use deep sequencing to gauge the effects of targeted gene knockouts on EV-mediated release of miRNAs on a whole genome scale. The method was applied in replicate screens, and its effectiveness is suggested by the identification of a handful of gene products previously implicated in EV biogenesis and/or release. In addition, at least one gene, ARHGEF18, was shown to be required for EV-mediated release of endogenous miRNAs.I find the method to be a clever implementation of miRNA and sgRNA processing pathways to link genetics with phenotype. The most important question is, does it work?

Thank you.

The identification of gene products previously identified as required for EV release suggests that it does.

We agree.

The approach did, however, require the authors to tackle a few technical bugs (eg, DNA sequencing, effects of different sgRNA-barcode combinations, etc) and it is not clear if further, unknown biases (if any) impact the utility of the method as a general method for evaluating the genetics of EV biogenesis.

We are very honest about limitations of the screen and explored some of them – it is genome wide, but we do not claim that it is a saturation screen.

A large number of genes not previously implicated in EV biology are asserted to play roles in EV biogenesis. At this stage, it is unclear if any of the 'new' hits play a direct role in EV cargo sorting, EV biogenesis, or EV release.

Our live cell pHluorin-CD63 TIRF experiments strongly suggest that MVE release is the problem in characterized hit-depleted cells – CD63-positive structures don’t appear to change, but their probability of release does.

As with most screens, the ultimate utility of the data will require substantial amount of follow-up work and will not be known for years to come.

We agree – thanks for acknowledging this.

Reviewer #2:In this manuscript Lu and colleagues perform a genome wide screen to get insights into the biogenesis and regulation of Extracellular Vesicle release.For this purpose, the authors set up an elegant system based on the generation of barcoded miRNAs that depict a sequence that has been previously shown to target miRNAs to multivesicular endosomes of T lymphocytes and in particular to their intralumenal vesicles (called exosomes once they are secreted). These findings from the group of Sanchez Madrid are largely exploited here (and would deserve better acknowledgement).

We did cite their paper and added additional acknowledgement.

The barcoded miRNAs are then used as reporters of EV release which is followed by a genome wide screen coupled with CRISPR/Cas9.The authors have performed an incredible amount of work, that certainly is a very interesting set up.There are several concerns with the manuscript, interpretation of results and the claims. Finally, one is a little left with the question on how such approach can really answer fundamental questions and pinpoint general or particular/specific machineries required for EV biogenesis and release.Essential revisions:1) EVs comprise very different types of vesicles that are either endosome derived (so called exosomes) and plasma membrane derived. One important issue in the field is the possibility in the future to ascribe functions to the different subpopulations. Several studies have described and reported distinct machineries for the biogenesis and release of exosomes and microvesicles that can be cell type and cargo specific (summarized in different recent reviews from different authors). Therefore, it is unclear what is the approach that is here developed bringing to the complexity that has been observed and described for the very different systems.

We have added new NTA Nanoparticle tracking which shows that our vesicles are rather uniform and small in size and more importantly, we added live cell imaging that now shows that our hits are really important for MVE exocytosis. These are very major new additions that tell us about the pathway involved, greatly increasing the significance of our story.

2) It is unclear from the claims in the manuscript and from the experiments done if one is trying to get hints on the biogenesis and release of endosome-derived exosomes or plasma membrane derived vesicles. It would certainly be important for the general audience to be clearer on this issue that is a real concern.

See above – our new data address this very well.

With this in mind, the characterization made, that is far from being very extensive, would seem to indicate that the authors maybe rather focus on endosome derived exosomes. This point should be consolidated better with additional proteins tested by Western blot, CD63, Tsg101. For example, CD63 should rather be used for the immunofluorescence experiments presented. Lamp1, if present in late endosomes, is much more enriched in lysosomes. The experiments presented in Figure 1 and Figure 6 are difficult then to interpret.

We have used CD63 is our exocytosis marker and we also now include CD63 IF (new Figure 1—figure supplement 2, Figure 5 and Figure 5—figure supplement 3.

Lamp1 is far from being the best "exosome" marker as in most cells Lamp1 is on the limiting membrane of the organelle and it is Lamp2 that is rather on the intraluminal vesicles. Therefore, it would be important to control whether in the cells under study Lamp1 is targeted to the intraluminal vesicles (or even if this is a consequence of the barcoded miRNAs?)

We added additional markers thus this is not needed at this stage (see above).

3) Following point 2, electron microscopy combined with extra immunofluorescence imaging would be more then appropriate and required to be sure, even to set background for the all findings, that the barcoded miRNAs, once expressed are not impacting in cell/organelle morphology and/or if they are targeted to the right place. This could also help understanding the effects of Rab27A which in most cells where it is expressed (not all cells express the "a" isoform) is present in Golgi TGN. Rab27a is localized to Lysosome Related organelles only in specialized cells. Therefore, is Rab27a in these manipulated cells recruited to multivesicular endosomes that may specialize after introduction of the barcoded miRs? For example, this is the case in HeLa cells expressing MHC Class II and all associated molecules for antigen presentation (Ostrowski et al., 2010). The sequence used has been described to my knowledge only in T cells, is it really driving the miRs to the right place in very different cells? In conclusion extra controls would support part of the claims. This would then help understanding what subpopulation of EVs is targeted in the study.

The live cell imaging really supports the idea that MVE release is what is impacted here. The NTA shows it is small vesicles. We have added more IF to show that the bEXOmiRs don’t change CD63compartment morphology (new Figure 1—figure supplement 2). As reported by others and shown here, Rab27 depletion decreases MVE exocytosis and EV recovery.

4) I would be more careful generalizing the concept based on the two cell lines used in this study although both express wnt.

We have modified the text as requested and toned down the Title.

5) The manuscript describes the involvement of Wnt signaling in the regulation of « EV biogenesis ». For me it is unclear what is really the mechanism of Wnt action: is it on the biogenesis per se or on the release/secretion step which are different parts of the pathway? It could be only on the release but whether there are biogenetic functions remains unclear. Are vesicles still formed? Is the cargo still loaded on vesicles? One has pretty much the tendency to put together biogenesis/release but the mechanisms involved in the generation and then in the release are very distinct. The manuscript should be clear on this point.

Again, we now use live cell imaging and show that loss of Rab27 blocks EV exocytosis and importantly, that GSK3 inhibition decreases Rab27 mRNA and protein. Thus, the pathway is now much clearer.

6) In subsection “Wnt signaling in EV biology” the authors test the effect of Wnt3a on EV release and show a decrease (is it on biogenesis or the release? by which mechanism?). This appears different from what was previously described in other cells such as microglia (Hooper et al., 2012), where Wnt3a induces exosome release, and inhibition of GSK3 does not altered exosome secretion, indicating that exosome release occurs through a GSK3-independent mechanism. Such discrepancy is not described, and the paper not quoted if I am not wrong.

We confirm here that different cells have different responses to GSK3 inhibition and have included this reference.

7) In concern to the effects of YKT6 that has been described by the group of Boutros as required for exosome-dependent Wnt release in Drosophila wing disc and cells and Hela cells, the paper of Gross et al., 2012 is not quoted.

We apologize and added this reference.

8) In Figure 1—figure supplement 1B and in the legend there is a discrepancy: in the figure it is 12.200g and in the legend 10.000 g and the time of not indicated in the figure.

Thank you for detecting that error-now corrected.

9) Figure 5B in the text there is a reference to cells versus EVs (subsection “A CRISPR/Cas9 screen using bEXOmiRs”) though unclear in the Figure.

We fixed this figure – they are now labeled more clearly at the top.

In conclusion, it is an elegant set up that deserves attention but certainly additional controls. At this point in my opinion it is more the set-up of a new method that maybe will set the stage for more meaningful experiments in the future. This may help to better target potential mechanisms and therefore understand the crosstalks and pathways influencing EV (rather subpopulations of EVs) biogenesis and release which should be separated as this are very different processes.

We thank the reviewer and wish to remind all the reviewers that this was submitted as a Resource.

Reviewer #3:In this manuscript, Albert Lu and colleagues, have designed a screen to identify regulators of the secretion of extracellular vesicles (EV). They have used a clever design by combining bar-coding of the EV with a single guide RNA to target genes using the Cas9 enzyme. Bar-coding is achieved using custom-designed miRNAs, the bEXOmiR.The authors used a pool screen format where an excess of cells is transfected with a complex library of viruses. Next generation sequencing then quantifies the abundance of specific miRNA in the supernatant of the pool of cells. Because each gene-specific guide RNA is combined with a specific bEXOmiR, inference can be drawn from the levels of a bEXOmiR in the supernatant.This is an elegant study that provides a valuable resource for researchers interested in EV. It also provides a proof of concept that such bar-coding of EV is feasible. Finally, it demonstrates elegantly that Wnt signaling is involved in regulating EV secretion.

Thank you.

In such pooled screens, the normalisation steps are really key to avoid mis-interpreting the data. In this case, Lu and colleagues have normalised first the bEXOmiR count to a cell line devoid of the nuclease Cas9. This should exclude any effect that the bEXOmiR barcode could have on packaging into EV or the effect the miRNAs could have on gene expression. The authors show convincingly that this is the case.

Thank you.

The next concern would be that the CRISPR targeted gene would affect the synthesis of the bar-coded miRNA instead of packaging and export in EV. These are admittedly two very different processes and it would be advantageous to distinguish them.As far as I could see, the authors did not use the comparison extra/intra cellular levels of bEXOmiR to sort their hits. It would be interesting to sort the hits based on whether there is an individual correlation between intracellular and extracellular amount of bEXOmiR upon gene knockout. This should be the case for the hits proposed to be involved in "miRNA processing and trafficking" (proposal based on previous annotations) (Figure 4). In theory it could help sort the other hits between miRNA processing and EV biosynthesis, especially for RNA binding proteins.

We have not sorted our hits according to packaging efficiency and note this in the revised text. We did address this in Figure 5 for the Nanostring experiment where specific miRNA levels only change in the EV but not cellular fractions. Orthogonal validation is thus important to ensure it was not a bEXOmiR synthesis issue - monitoring the release of endogenous miRNAs instead of bEXOmiRs and creating CRISPR cell lines that show the same phenotype for specific hits without any bEXOmiRs. By using 10 different bEXOmiRs for every gene, we tried to minimize any biosynthesis bias. In addition, we now more clearly explain Figure 3 where we show no correlation between abundance of total bEXOmiRs in cells compared with EVs. We also validated the roles of the hits using TIRF, confirming the use of these reporters to good end.

The final concern with screen design would be whether some genes could affect the uptake and degradation of EV. Indeed, as this is a suspension culture, cells have in theory the capacity to affect the extracellular levels of EV also through endocytosis and degradation. A hit that would affect the membrane surface of EV and thus affect EV binding to neighbouring cells after secretion could, in theory affect the overall levels of EV without affecting their synthesis and release. I wonder if this could be an explanation for the effect of some of the extracellular proteins identified (Figure 4). I wonder if the authors can exclude this effect or control for it?

Although endocytosis of EVs by surface proteins might lead to a decrease in EV bEXOmiR abundance, we did not detect a large cluster of endocytosis genes in our screen and state this in the text.

" Unexpectedly, YKT6, which appeared as a suppressor in the initial screen (Figure 2B), was uncovered as a slight activator instead; this may be related to differences in the bEXOmiRs associated with this gene in the two separate analyses..… This discrepancy highlights the importance of orthogonal hit validation experiments."It also highlights the difficulty in interpreting the significance score (-Log10(P-value) to evaluate reproducibility. I have trouble evaluating how many pairs of bEXOmiRs-sgRNA gave similar results with this metric.

Our screen has many limitations and the CASTLE score emphasizes the outliers. Again, different bEXOmiRs have variable efficiencies as do different guides, and one would need to redo the screen with a more limited, enhanced set to have better genome-wide coverage. We have added better data on recovery of each bEXOmiR for all the hits in a new Table 7 as requested.

It would be surprising that the YKT6 case would occur if the result found in the main screen is attributable to 3 or 4 pairs of bEXOmiR+sgRNA. I think it would help clarify the data if the authors provided a table with the result for each gene: with how many constructs gave a result and what score.

Done as requested – usually 2-3 guide-bEXOmiR associations supported each hit. Text modified.